# Phase transitions above the upper critical dimension

Bertrand Berche[a,d], Tim Ellis[b,d], Yurij Holovatch[c,d,b], Ralph Kenna[b,d]

**a** Laboratoire de Physique et Chimie Théoriques,
Université de Lorraine - CNRS, UMR 7019, Nancy, B.P. 70239, F-54506 Vandœuvre les Nancy, France
**b** Centre for Fluid and Complex Systems,
Coventry University, Coventry, CV1 5FB, United Kingdom
**c** Institute for Condensed Matter Physics,
National Acad. Sci. of Ukraine, UA–79011 Lviv, Ukraine
**d** $\mathbb{L}^4$ Collaboration & Doctoral College for the Statistical Physics of Complex Systems,
Leipzig-Lorraine-Lviv-Coventry, Europe

April 14, 2022

## Abstract

**These lecture notes provide an overview of the renormalization group (RG) as a successful framework to understand critical phenomena above the upper critical dimension $d_{\mathrm{uc}}$. After an introduction to the scaling picture of continuous phase transitions, we discuss the apparent failure of the Gaussian fixed point to capture scaling for Landau mean-field theory, which should hold in the thermodynamic limit above $d_{\mathrm{uc}}$. We recount how Fisher's dangerous-irrelevant-variable formalism applied to thermodynamic functions partially repairs the situation but at the expense of hyperscaling and finite-size scaling, both of which were, until recently, believed not to apply above $d_{\mathrm{uc}}$. We recall limitations of various attempts to match the RG with analytical and numerical results for Ising systems. We explain how the extension of dangerous irrelevancy to the correlation sector is key to marrying the above concepts into a comprehensive RG scaling picture that renders hyperscaling and finite-size scaling valid in all dimensions. We collect what we believe is the current status of the theory, including some new insights and results. This paper is in grateful memory of Michael Fisher who introduced many of the concepts discussed and who, half a century later, contributed to their advancement.**

# 1  Prelude: Definitions and notations

Without loss of generality, we present some of this exposition within the framework of the Ising model as it is the original and most fundamental spin model and the one generically used in textbooks. Generalization to other models is straightforward and implicit. Notwithstanding this, we find it illustrative to present most of the work in the context of $\phi^n$ theory. For clarity, we first specify the notations used and some specific vocabulary. This section can be skipped by readers well versed in statistical physics theories of critical phenomena.

The Ising model is defined on a hypercubic lattice $\Lambda$, of volume $\ell^d$, and edge length $\ell = La$ in space of dimension $d$. Here $L$ is a number of unit cells in any direction of the lattice and $a$ is their individual length. The sites of $\Lambda$ are assigned coordinates $\mathbf{x}_i = (x_{i_1}, x_{i_2}, \ldots, x_{i_d})$ with $x_{i_n} = 1, 2, \ldots, L$. At each lattice site sits a dimensionless Ising spin variable $s_{\mathbf{x}_i} = \pm 1$. In the case of short range interactions, these local degrees of freedom interact between nearest neighbours, $(\mathbf{x}_i, \mathbf{x}_j)$ through the so-called "exchange interaction", as well as with an external magnetic field, so that the Hamiltonian is

$$\mathcal{H} = -\sum_{(\mathbf{x}_i, \mathbf{x}_j)} J s_{\mathbf{x}_i} s_{\mathbf{x}_j} - \sum_{\mathbf{x}_i} H s_{\mathbf{x}_i}. \tag{1}$$

Here, the interaction coupling $J$ and the magnetic field $H$ have dimensions of energy. They are considered uniform for the simplest version of the model presented here. Generalisation to long-range models is straightforward.

This Ising model above $d = 1$ exhibits a phase transition between an ordered state at low temperature and a disordered state at high temperature. At the transition, which occurs at finite temperature, the physical properties become singular and this is what we are interested in. An equilibrium order parameter discriminates between the two phases. It is zero above the transition in the thermodynamic limit and non-zero below. Thermodynamic properties such as these are deduced from the partition function

$$Z(\beta, H) = \sum_{\{s_{\mathbf{x}_i}\}} e^{-\beta \mathscr{H}} \tag{2}$$

where the sum is over all the spin configurations and $\beta$ is the inverse temperature. The free energy is

$$F_\infty(\tau, h) = -\beta^{-1} \ln Z_\infty(\beta, H) \tag{3}$$

where $\tau = (\beta J - \beta_c J)/(\beta_c J)$ and $h = \beta H$ with $\beta_c$ the inverse critical temperature. The subscript $\infty$ here, and throughout, indicates that the observable is considered in the thermodynamic limit. Various thermodynamic quantities are obtained through derivatives of the free energy density $f_\infty = V^{-1} F_\infty$ wrt the temperature and/or the magnetic field. (Here $V = L^d$.) E.g. the magnetization, internal energy, susceptibility and specific heat, are

$$m_\infty(\tau, h) = \frac{\partial f_\infty(\tau, h)}{\partial h}, \tag{4}$$

$$e_\infty(\tau, h) = \frac{\partial f_\infty(\tau, h)}{\partial \tau}, \tag{5}$$

$$\chi_\infty(\tau, h) = \frac{\partial^2 f_\infty(\tau, h)}{\partial h^2}, \tag{6}$$

$$c_\infty(\tau, h) = \frac{\partial^2 f_\infty(\tau, h)}{\partial \tau^2}. \tag{7}$$

In these definitions, dimensional factors (and signs) are not considered as they are not needed for our purpose.

In practical numerical simulations, the equilibrium properties of the model are defined after averaging with appropriate weights. E.g. for a physical quantity $Q$, defined in terms of the local degrees of freedom,

$$\langle Q \rangle = \frac{1}{Z} \sum_{\{s_{\mathbf{x}_i}\}} Q(\{s_{\mathbf{x}_i}\}) e^{-\beta \mathscr{H}}. \tag{8}$$

Numerically, the sum over spin configurations is sampled via a random walk in configuration space and the weighted average is produced by specific transition rules among the configurations sampled. These transition rules and the detail of the building of the configurations are properly defined for each type of algorithm (Metropolis, cluster algorithms, reweighting algorithms, Fukui-Todo algorithms, etc.). This results usually in a list of values for the average "spin" and average energy,

$$S = \sum_{\mathbf{x}_i} s_{\mathbf{x}_i}, \qquad E = \sum_{(\mathbf{x}_i, \mathbf{x}_j)} s_{\mathbf{x}_i} s_{\mathbf{x}_j}, \tag{9}$$

so that for each configuration $S$ and $E$ take values $S_n$, $E_n$ viz

$$
\begin{array}{ccc}
n & S_n & E_n \\
1 & S_1 & E_1 \\
2 & S_2 & E_2 \\
3 & S_3 & E_3 \\
\vdots & \vdots & \vdots
\end{array}
\tag{10}
$$

The equilibrium value of an observable $Q(S, E)$ is then calculated by averaging the values in the list,

$$
\langle Q \rangle = \frac{1}{N_{\text{conf}}} \sum_{n=1}^{N_{\text{conf}}} Q(S_n, E_n).
\tag{11}
$$

Here, $N_{\text{conf}}$ is the number of useful Monte Carlo (MC) iterations after thermalization and after other possible transient regimes have been discarded. In order to simplify later definitions, we denote the average over the MC iterations with brackets:

$$
\langle (\dots) \rangle = \frac{1}{N_{\text{conf}}} \sum_{n=1}^{N_{\text{conf}}} (\dots).
\tag{12}
$$

The equilibrium spontaneous magnetization (in zero magnetic field) requires special precaution. The values of $S_n$ have the same probability to be positive or negative. This is because of the invariance of the model under $\mathbb{Z}_2$ symmetry which sends $S_{\mathbf{x}_i} \to -S_{\mathbf{x}_i} \forall i$. The non-zero value of the equilibrium spontaneous magnetization in a real material results from spontaneous symmetry breaking in the thermodynamic limit $L \to \infty$. In MC simulations of finite systems, the symmetry must be artificially broken and this is usually done by using absolute values. The equilibrium spontaneous magnetization (density) for a system of size $L$ is therefore defined as

$$
m_L = L^{-d} \langle |S_L| \rangle.
\tag{13}
$$

The susceptibility is defined by the corresponding second moment,

$$
\chi_L = L^{-d} (\langle S_L^2 \rangle - \langle |S_L| \rangle^2).
\tag{14}
$$

The subtracted term here is needed only in the ordered phase to get rid of the non-zero average magnetization there and delivers a vanishing susceptibility in the limit $T \to 0$. The energy and specific heat are defined accordingly:

$$
e_L = L^{-d} \langle E_L \rangle,
\tag{15}
$$

$$
c_L = L^{-d} (\langle E_L^2 \rangle - \langle E_L \rangle^2).
\tag{16}
$$

We use the term "magnetic sector" to refer to quantities defined solely in terms of the $S_n$'s and we use the term "energy sector" for those defined solely in terms of the $E_n$'s. Quantities in the energy sector are often more difficult to analyze due to the presence of strong non-singular contributions at the transition.

In establishing the list over MC iterations, we may need more information (for example the individual values of the local spins for each configuration) if local quantities are to be calculated. This is the case if we want to measure the order parameter profile:

$$
m(\mathbf{x}_i) = \langle |s_{\mathbf{x}_i}| \rangle.
\tag{17}
$$

The (connected) correlation function and the correlation length are defined from this local order parameter,

$$g_L(\mathbf{x}, \tau, h) = \langle m(0, t, h) m(\mathbf{x}, \tau, h) \rangle - \langle m(0, \tau, h) \rangle \langle m(\mathbf{x}, \tau, h) \rangle \sim e^{-|\mathbf{x}|/\xi_L(\tau, h)}. \qquad (18)$$

It is useful to define local quantities, like $s_{\mathbf{x}_i}$, in terms of Fourier modes $\psi_{\mathbf{k}}$,

$$s_{\mathbf{x}_i} = \sum_{\mathbf{k} \in \mathcal{K}} \tilde{s}_{\mathbf{k}} \psi_{\mathbf{k}}. \qquad (19)$$

The "single-mode equilibrium magnetization" and the corresponding susceptibility are thus

$$m_{\mathbf{k}} = \langle |\tilde{s}_{\mathbf{k}}| \rangle, \qquad (20)$$
$$\chi_{\mathbf{k}} = \langle |\tilde{s}_{\mathbf{k}}|^2 \rangle - \langle |\tilde{s}_{\mathbf{k}}| \rangle^2. \qquad (21)$$

We denote by $\mathcal{K}$ the set of all $\mathbf{k}$-modes. If $m(\mathbf{x})$ possesses certain symmetries we can partition the modes in two sets, $\mathcal{K} = \mathcal{Q} \cup \mathcal{G}$ where the modes pertaining to $\mathcal{Q}$ possess the same symmetry as $m(\mathbf{x})$ and $\mathcal{G}$ is the orthogonal set. Then, we refer to the modes which mostly contribute to Eq. (17), $\mathbf{k} \in \mathcal{Q}$, as $Q$-modes. The modes in $\mathcal{G}$ are called $G$-modes. The reason for this notation will become clear later on. Suffice to say for now that $G$ refers to "Gaussian" and $Q$ refers to a seemingly inert parameter that turns out to play a crucial role in understanding critical phenomena in high dimensions.

In a system with periodic boundary conditions (PBC) the Fourier modes are plane waves

$$\psi_{\mathbf{k}} = L^{-d/2} e^{i\mathbf{k} \cdot \mathbf{x}} \qquad (22)$$

and the boundary conditions select the wave vectors $\mathbf{k} = (2\pi/L)\mathbf{n}$, $\mathbf{n} \in \mathbb{Z}^d$. Translation invariance requires a uniform equilibrium profile $m(\mathbf{x}) = $ constant, and only the zero mode for which $\mathbf{n} = 0$ has the symmetry of the uniform profile. It follows that

$$m(\mathbf{x}) = m_0 = \langle |\tilde{s}_{\mathbf{k}=0}| \rangle \qquad (23)$$

and $\mathcal{Q}$ contains only the zero mode while $\mathcal{G}$ contains all the other modes. A typical element of $\mathcal{G}$ is for example $\mathbf{k}_G = (2\pi/L)(1, 0, 0, \dots, 0)$. Although $\mathbf{k}_G$ does not contribute to the equilibrium magnetization, it is interesting in some circumstances to analyze the behaviour of quantities like $m_{\mathbf{k}_G}$.

In a system with free boundary conditions (FBC), the Fourier modes are sine waves

$$\psi_{\mathbf{k}} = \sqrt{2/L} \prod_{\alpha=1}^{d} \sin k_\alpha x_\alpha \qquad (24)$$

with wave vectors $k_\alpha = n_\alpha \pi/(L+1)$, $n_\alpha = 1, 2, \dots, L$. The equilibrium profile $m(\mathbf{x})$ is an even function wrt the center of the system, and only the modes for which the $n_\alpha$'s are all odd share this symmetry. These are the $Q$-modes, a typical representative of which is $\mathbf{k}_Q = (\pi/(L+1))(1, 1, 1, \dots, 1)$. All the other modes are $G$-modes and a typical $G$-mode is $\mathbf{k}_G = (\pi/(L+1))(2, 1, 1, \dots, 1)$. There is no zero mode for FBC's.

Having set up this framework, and established the notation, we next proceed to introduce the problem at hand — scaling for critical phenomena in high dimensions.

# 2 Introduction: Second-order phase transitions and the scaling picture

In this paper, we want to revisit an old question, namely scaling in high dimensions and the role of dangerous irrelevant variables (DIV) — a very smart stratagem introduced by Michael Fisher in the 1980's. Fisher introduced the concept to reconcile the celebrated renormalization group (RG) picture of critical phase transitions with results from mean-field theory (MFT). The latter holds above the upper critical dimension where fluctuations are known to become irrelevant. We are not aware of any journal article by Fisher on this problem, except a short exposition in Ref. [1], but we can find the argument developed in the wonderful appendix D of a course that he gave in Stellenbosch in 1982 [2]. There he concluded by saying

> *The moral of this story is that the standard scaling relations for critical exponents depend, in their derivation, on assumptions, usually left tacit, about the nonsingular or nonvanishing behaviour of various scaling functions and their arguments. In many cases these assumptions are valid and may be confirmed by explicit calculation (or other knowledge) but in certain circumstances they may fail, in which case an exponent relation may change its form. Other nontrivial cases of dangerous irrelevant variables are known so that phenomenon, although not common, is not truly exceptional.*

Earlier, in the same appendix, he made the important statement:

> *Notice [. . . ] that the renormalization group framework has been preserved intact: the only flaw in the original argument was a failure to recognize and allow for possible singular behaviour of the scaling function.*

We revisit these statements at the end of this paper and adapt them for our main conclusions which are not dissimilar. In a nutshell, what Fisher and subsequent authors did for free energy and its derivatives, we do for the correlation function and correlation length. This is needed for the RG framework to be "preserved intact" for finite-size systems too.

We consider various systems [e.g., $O(N)$ models, percolation, and tricritical points] with PBC's, for which we believe that the theory is fully settled. After that, we address the case of FBC's (Section 10). There, the results as we will see are still incomplete.

These notes should not be considered as a review. We do not purport to give a full account of the vast literature published over almost half a century, but we will refer to those papers that we consider as being the most important in the field. Several years ago, we published a contribution in the form of a review [3] and the present paper has a different purpose. It aims at presenting advances in our understanding of scaling above the upper critical dimension over the past hundred years with the progress and setbacks inherent to the evolution of science.

So, the context is that of second-order phase transitions. Let us recall some of the essential results of the theory of phase transitions and critical phenomena. According to the standard picture, six critical exponents, $\alpha$, $\beta$, $\gamma$, $\delta$, $\nu$ and $\eta$, to follow nomenclature coined by Fisher[1], describe the singularities of the "major" physical quantities when approaching the critical

---

[1]Fisher standardised definitions and formulas for critical point singularities in 1966 [4].

point $h = 0$, $t \to 0^{\pm}$, $t = 0$, $h \to 0^{\pm}$, namely:

$$c_{\infty}(t) \simeq \frac{A^{\pm}}{\alpha}|t|^{-\alpha}, \tag{25}$$

$$m_{\infty}(t) \simeq B^{-}|t|^{\beta}, \ t < 0, \qquad m_{\infty}(h) \simeq D_c|h|^{1/\delta}, \tag{26}$$

$$\chi_{\infty}(t) \simeq \Gamma^{\pm}|t|^{-\gamma}, \tag{27}$$

$$\xi_{\infty}(t) \simeq \xi^{\pm}|t|^{-\nu}, \tag{28}$$

and that of the correlation function right at the critical point,

$$g_{\infty}(t = 0, h = 0, \mathbf{x}) \sim \frac{1}{|\mathbf{x}|^{d-2+\eta}}. \tag{29}$$

In these expressions, $t$ and $h$ measure the distance to the critical point, $t = T - T_c$ and $h = H - H_c$ (with usually $H_c = 0$).[2] As stated, the subscript indicates the thermodynamic limit, where the typical size $L$ (hence number of interacting degrees of freedom) tends to infinity. The symbols "$\simeq$" and "$\sim$" in equations (25)–(29), although widely used in the context of critical phenomena, may require some clarification. They are in no way to be understood as the same as the equality sign "$=$". The first symbol is usually used with the meaning of an equivalence or first approximation between functions, while $\sim$ refers to a similarity between functions. We could say that $c_{\infty}(t) \simeq \frac{A^{\pm}}{\alpha}|t|^{-\alpha}$, and $c_{\infty}(t) \sim |t|^{-\alpha}$, but each of these are in no way complete — tradition holds that they imply there is something more. It is to be understood that upon this *singular* behaviour described by the critical exponent $\alpha$, there could be a bunch of regular terms (e.g. $D_0 + D_1|t| + \dots$). It should also be understood that the leading singular behaviour itself may need to be completed by correction terms called corrections to scaling which may play a role further away from the transition temperature. The expression for $c_{\infty}(t)$ then may take a form like [7]

$$
\begin{array}{llll}
c_{\infty}(t) & = & D_0^{\pm} + D_1^{\pm}|t| + \dots & \text{regular background} \\
& + & \frac{A^{\pm}}{\alpha}|t|^{-\alpha}[1+ & \text{leading singularity} \\
& & + \, a_1^{\pm}|t|^{\omega} + a_2^{\pm}|t|^{2\omega} + \dots & \text{leading corrections} \\
& & + \, a'_1|t|^{\omega'} + a'^{\pm}_2|t|^{2\omega'} + \dots & \text{next corrections} \\
& & + \, b_1^{\pm}|t| + b_2^{\pm}|t|^2 + \dots] & \text{analytic corrections} \\
& \times & (-\ln|t|)^{\hat{\alpha}} \times (1 + \dots). & \text{logarithmic corrections.}
\end{array}
$$

The amplitudes $A^{\pm}$ and exponent $\alpha$ are associated with the leading singularity. The corrections to scaling due to irrelevant fields are denoted as $a_n^{\pm}$ and $\omega$, $a'^{\pm}_n$ and $\omega'$, .... There might be also analytic corrections associated to non-linearities of the relevant scaling fields (here $b_n^{\pm}$). Eventually, multiplicative logarithmic corrections (here $\hat{\alpha}$) may be observed. These corrections may have different origins and are discussed in Refs. [8–11]. They can occur in particular right at the upper critical dimension.

---

[2]For a discussion of the quality of scaling in terms of the variable $t = T - T_c$ and in terms of $\tau = \beta_c - \beta$, with $\beta = (k_B T)^{-1}$, see the work of Ian Campbell and colleagues, e.g. [5,6]. Here we conform to the most standard choice, but we stress that $\beta_c - \beta$ is to be preferred in numerical estimates.

This being said, four scaling relations exist among the six standard critical exponents,

$$\nu d = 2 - \alpha, \tag{30}$$

$$\alpha + 2\beta + \gamma = 2, \tag{31}$$

$$\beta(\delta - 1) = \gamma, \tag{32}$$

$$\nu(2 - \eta) = \gamma, \tag{33}$$

and, as a consequence, the knowledge of two of them is enough to derive all the others.

The first of these was developed in 1965 by Widom [12–14] and later by Kadanoff [15]. The second was originally proposed in 1963 by Essam and Fisher [16], the third a year later by Widom [17] and the fourth by Fisher [18]. The first equality in this list is often referred to as Josephson's scaling relation but Josephson's contribution was actually an associated inequality ($\nu d \geq 2 - \alpha$) derived a couple of years later [19]. Likewise there is an inequality related to the second formula ($2\beta + \gamma \geq 2 - \alpha$) which was rigorously proven by Rushbrooke in 1963 [20]. There is also an inequality ($\beta(\delta - 1) \leq \gamma$) associated with the third relation – proved in 1967 by Griffiths [21]. An inequality associated with Fisher's scaling relation was proved a few years later [22, 23]. The main focus of this paper is the first scaling relation. Because it involves dimensionality, it is called the *hyperscaling* relation. It is frequently said to *fail* in high dimensions where $\alpha$ and $\nu$ adhere to their mean-field values. In Section 9 we will present a new form for it, valid in all dimensions. This new format necessitates a new critical exponent which will emerge from the scaling exponents as the other do but in a less obvious manner.

A fundamental hypothesis in the theory of critical phenomena is called the *homogeneity assumption* for the singular part of the free energy density [24, 25]. It relies entirely on the RG and states that $f_\infty^{\text{sing}}$ is a generalized homogeneous function of the arguments $t$ and $h$, the relevant thermal and magnetic fields already introduced, with corresponding RG eigenvalues $y_t$ and $y_h$:

$$f_\infty^{\text{sing}}(t, h) = b^{-d} f_\infty^{\text{sing}}(b^{y_t} t, b^{y_h} h). \tag{34}$$

Here, $d$ is the space dimension and $b$ is an arbitrary rescaling factor, This equation is often rewritten as

$$f_\infty^{\text{sing}}(t, h) = |t|^{d/y_t} \mathscr{F}^{\pm}(h|t|^{-y_h/y_t}) \tag{35}$$

where $\mathscr{F}^{\pm}(y) = f_\infty^{\text{sing}}(\pm 1, y)$ are two universal functions (above and below the critical temperature), called scaling functions, of the unique argument $y = h|t|^{-y_h/y_t}$ and defined by (34) with the choice of rescaling factor $b = |t|^{-1/y_t}$. We will use calligraphic upper case letters to denote such scaling functions when the dependence with some variables is omitted if these variables keep zero or constant values.

The homogeneity assumption is complemented by a similar hypothesis concerning the correlation function and the correlation length,

$$
\begin{aligned}
g_\infty(\mathbf{x}, t, h) &= b^{-2x_\phi} g_\infty(b^{-1}\mathbf{x}, b^{y_t} t, b^{y_h} h) \\
&= |\mathbf{x}|^{-2x_\phi} \mathscr{G}_{\mathbf{u}}(|\mathbf{x}|/|t|^{-1/y_t}, |\mathbf{x}|/|h|^{-1/y_h}), \text{ with } b = |\mathbf{x}|. \tag{36} \\
\xi_\infty(t, h) &= b \xi_\infty(b^{y_t} t, b^{y_h} h) \\
&= |t|^{-1/y_t} \Xi^{\pm}(h|t|^{-y_h/y_t}), \text{ with } b = |t|^{-1/y_t}, \tag{37} \\
&\text{or } |h|^{-1/y_h} \tilde{\Xi}^{\pm}(t|h|^{-y_t/y_h}), \text{ with } b = |h|^{-1/y_h}. \tag{38}
\end{aligned}
$$

Here, $\mathbf{u} = \mathbf{x}/|\mathbf{x}|$, but because we will only focus on isotropic critical phenomena, this vector subscript in the scaling function of the correlation function is unnecessary and will be omitted in the rest of the paper. The explicit definitions here are for the order parameter correlation function and $x_\phi$ is the corresponding order parameter scaling dimension which we will discuss later. A thermal correlation function could be similarly defined.

The critical behaviours of the magnetization, susceptibility, internal energy and specific heat then follow by taking the appropriate derivatives of (34) wrt $t$ or $h$,

$$m_\infty(t, h) = b^{-d+y_h} m_\infty(b^{y_t} t, b^{y_h} h), \tag{39}$$

$$\chi_\infty(t, h) = b^{-d+2y_h} \chi_\infty(b^{y_t} t, b^{y_h} h), \tag{40}$$

$$e_\infty(t, h) = b^{-d+y_t} e_\infty(b^{y_t} t, b^{y_h} h), \tag{41}$$

$$c_\infty(t, h) = b^{-d+2y_t} c_\infty(b^{y_t} t, b^{y_h} h). \tag{42}$$

The definition of the critical exponents according to the standard terminology [4] then follows from the elimination of the $x = b^{y_t} t$ or $y = b^{y_h} h$ dependencies. *At the critical temperature,* the choice $b = |h|^{-1/y_h}$, gives

$$m_\infty(0, h) \simeq D_c^{-1/\delta} |h|^{1/\delta}, \quad \delta = \frac{y_h}{d - y_h}, \quad D_c^{-1/\delta} = m_\infty(0, 1). \tag{43}$$

In zero magnetic field, $b = |t|^{-1/y_t}$ delivers

$$m_\infty(t, 0) \simeq B^- |t|^\beta, \quad \beta = \frac{d - y_h}{y_t}, \quad B^- = m_\infty(-1, 0), \; t < 0, \tag{44}$$

$$\chi_\infty(t, 0) \simeq \Gamma^\pm |t|^{-\gamma}, \quad \gamma = \frac{2y_h - d}{y_t}, \quad \Gamma^\pm = \chi_\infty(\pm 1, 0), \tag{45}$$

$$c_\infty(t, 0) \simeq \frac{A^\pm}{\alpha} |t|^{-\alpha}, \quad \alpha = \frac{2y_t - d}{y_t}, \quad \frac{A^\pm}{\alpha} = c_\infty(\pm 1, 0). \tag{46}$$

The same strategy operates in (37) to give

$$\xi_\infty(t, 0) \simeq \Xi^\pm |t|^{-\nu}, \quad \nu = \frac{1}{y_t}, \quad \Xi^\pm = \xi_\infty(\pm 1, 0), \tag{47}$$

$$\xi_\infty(0, h) \simeq \tilde{\Xi}^\pm |h|^{-\nu_c}, \quad \nu_c = \frac{1}{y_h}, \quad \tilde{\Xi}^\pm = \xi_\infty(0, \pm 1), \tag{48}$$

and in equation (36), the choice $b = |\mathbf{x}|$, together with the isotropy requirement, leads at criticality $t = h = 0$ to

$$d - 2 + \eta = 2x_\phi. \tag{49}$$

It is now clear that the picture in Eqs.(25)–(28) is not complete. Using the notations of

| Symmetry | models | 2d | 3d | 4d | 5d | 6d | 7d | ... |
|---|---|---|---|---|---|---|---|---|
| $Z_q$ | | | | | | | | |
| $Z_1$ | Percolation | 2d Perco | 3d Perco | 4d Perco | 5d Perco | $\phi^3$ + logs | $\phi^3$ | |
| $q = 2$ | Ising, binary alloys, ... | 2d IM | 3d IM | $\phi^4$ + logs | $\phi^4$ | $\phi^4$ | $\phi^4$ | |
| $q = 3$ | Potts $q = 3$, surface adsorption, ... | 2d $Z_3$ | 1st | 1st | 1st | 1st | 1st | |
| $q = 4$ | Potts $q = 4$, Ashkin-Teller, surface adsorption, ... | 2d $Z_4$ | 1st | 1st | 1st | 1st | 1st | |
| $O(N)$ | | | | | | | | |
| $N \to 0$ | SAW, | 2d SAW | 3d SAW | $\phi^4$ + logs | $\phi^4$ | $\phi^4$ | $\phi^4$ | |
| $N = 2$ | XY, superconductivity, BEC, ... | KT | 3d $O(2)$ | $\phi^4$ + logs | $\phi^4$ | $\phi^4$ | $\phi^4$ | |
| $N = 3$ | Heisenberg, | No order | 3d $O(3)$ | $\phi^4$ + logs | $\phi^4$ | $\phi^4$ | $\phi^4$ | |

Table 1: Examples of universality classes. 1st denotes first-order phase transitions. Perco, IM, SAW and KT stand for percolation, the Ising model, self-avoiding walk and the Kosterlitz-Thouless transition. Above a certain space dimension, mean-field theory (here denoted as $\phi^n$) describes properly the critical properties.

Refs. [26] for the critical amplitudes one expands it to

$$h = 0, \ t \to 0^{\pm}, \qquad t = 0, \ h \to 0^{\pm},$$

$$f_{\infty}^{\mathrm{sing}}(t,0) \simeq F^{\pm}|t|^{2-\alpha}, \qquad f_{\infty}^{\mathrm{sing}}(0,h) \simeq F_c|h|^{1+1/\delta}, \tag{50}$$

$$m_{\infty}(t,0) \simeq B^{-}|t|^{\beta}, \qquad m_{\infty}(0,h) \simeq D_c^{-1/\delta}|h|^{1/\delta}, \tag{51}$$

$$e_{\infty}(t,0) \simeq \frac{A^{\pm}}{\alpha(1-\alpha)}|t|^{1-\alpha}, \qquad e_{\infty}(0,h) \simeq E_c|h|^{\epsilon_c}, \tag{52}$$

$$\chi_{\infty}(t,0) \simeq \Gamma^{\pm}|t|^{-\gamma}, \qquad \chi_{\infty}(0,h) \simeq \Gamma_c|h|^{1/\delta-1}, \tag{53}$$

$$c_{\infty}(t,0) \simeq \frac{A^{\pm}}{\alpha}|t|^{-\alpha}, \qquad c_{\infty}(0,h) \simeq \frac{A_c}{\alpha_c}|h|^{-\alpha_c}, \tag{54}$$

$$m_{T\infty}(t,0) \simeq m^{\pm}|t|^{\beta-1}, \qquad m_{T\infty}(0,h) \simeq m_c|h|^{\epsilon_c-1}, \tag{55}$$

$$\xi_{\infty}(t,0) \simeq \xi^{\pm}|t|^{-\nu}, \qquad \xi_{\infty}(0,h) \simeq \xi_c|h|^{-\nu_c}. \tag{56}$$

The cross derivative of the free energy density wrt $t$ and $h$, i.e., the magnetocaloric coefficient $m_T$, is often measured in experiments — since it is more singular that the magnetization itself — but its critical exponent is not an independent quantity. The standard definitions of some exponent combinations which occur above are: $\alpha_c = \alpha/(\beta\delta), \beta_c = \beta/(\beta\delta), \gamma_c = \gamma/(\beta\delta),$ $\nu_c = (\delta - 1)/(2\delta) =, \epsilon_c = 1 - \alpha_c$. This makes all together a collection of critical exponents, with many redundancies, the most commonly used being the six that we mentioned initially.

Second-order phase transitions may then be classified according to universality classes in which physical systems share the same set of critical exponents. These universality classes essentially depend on the space dimensionality and the symmetries of the order parameter. They also depend on the range of interactions, a situation discussed in Section 8. A few universality classes are given in Table 1. What we discuss in these lectures describes the situation which holds everywhere in the table where a $\phi^n$ appears.

At this point we have sketched the general scaling picture based on a homogeneity assumption for the free energy, correlation function and correlation length. The six main critical exponents $\alpha$, $\beta$, $\gamma$, $\delta$, $\nu$ and $\eta$, together with cross-derivative exponents, are each derivable from the scaling dimensions $y_t$, $y_h$ and $x_\phi$ together with the dimensionality $d$. There are four scaling relations, one of which involves $d$. We have not yet given values of the critical exponents or values of the scaling dimensions. In the next section we do the former and in the following section we derive the latter in circumstances where order-parameter fluctuations might be expected to be negligible. The incompatibility of the two will expose the failure identified by Fisher at the start of this section.

## 3 Ginzburg-Landau mean-field theory

Ginzburg-Landau theory is a mean-field theory (MFT), i.e. one which does not take into account the critical fluctuations of the order parameter. It is based on simple assumptions which essentially boil down to writing the free energy density as a power expansion of the order parameter and its derivatives. This can be applied to many different systems. A simplified version in which the order parameter is spatially uniform is called the Landau theory.

Like any theory with a universal ambition, it cannot be accurate in its details for every individual circumstance. One can hardly expect a theory that is able to describe extremely different systems (e.g. liquid-gas transition, superconductivity or ferromagnetism) to at the same time provide exact results for each of these systems. Nevertheless, being based on minimal hypotheses, Landau theory is an instructive approach and is generally qualitatively correct. Furthermore, Landau theory is accurate in most of its predictions in high dimensional systems, for which fluctuations cease to play the major role. However, the minimal, instructive and phenomenological features of the model are in no way shortcomings or failings. In trimming back to bare essentials, the theory explores the roles of fundamental concepts such as space dimensionality and order-parameter symmetries for universal critical phenomena, without worrying about the quirks of individual physical set-ups.

This theory is covered in virtually any course on phase transitions and critical phenomena, so that the following cursory exposition is probably superfluous for most readers. But in order to be complete and to further fix the notations, we will sketch the main lines of Ginzburg-Landau theory. Let us assume that a physical system is described at thermal equilibrium by the set of equations

$$Z = \int D\phi \, e^{-F[\phi]}, \tag{57}$$

$$F[\phi] = \int d^dx \, f(\phi, \boldsymbol{\nabla}\phi), \tag{58}$$

$$f(\phi, \boldsymbol{\nabla}\phi) = \tfrac{1}{2}r\phi^2(\mathbf{x}) + \tfrac{1}{3}u_3\phi^3(\mathbf{x}) + \tfrac{1}{4}u_4\phi^4(\mathbf{x}) + \tfrac{1}{6}u_6\phi^6(\mathbf{x}) - h\phi(\mathbf{x}) + \tfrac{1}{2}|\boldsymbol{\nabla}\phi|^2. \tag{59}$$

Here, the partition function $Z$ is a functional integral over the values of $\phi(\mathbf{x})$, a real scalar field that we call the matter field to distinguish it from external fields (temperature, magnetic field, . . . ). $F[\phi]$ is the free energy, a functional of the same field, and $f$ is a free energy density. The limitation that our presentation here is in terms of a scalar matter field (i.e., with $O(1)$ symmetry) is for the simplicity of notation only; it does not have severe consequences for the overall concepts and vector fields would only describe $O(N)$ theories with higher values of $N$.

We obviously borrow the denomination of matter field from classical field theories for which one would call $F[\phi]$ an action and $f$ a Lagrangian density.

The coefficient $r \sim t$ and, for our purposes, we take $u_n \geq 0$ for all $n$, although the study of first-order phase transitions would partially relax this condition. The coefficient of the highest power must anyway be positive to guarantee stability for a finite equilibrium value of the order parameter. More generically, $r$ controls the phase transition and must be positive in the phase for which the equilibrium order parameter vanishes and negative in the ordered phase. Usually, we think about this parameter in terms of the distance of the temperature to its critical value $T_c$, but $t$ may also be another type of parameter measuring the distance to some critical value (in percolation theory for example, there is no temperature). Again, we frame our discussion it terms of temperature for expediency and without losing this generality. An external magnetic field is essentially represented by $h$ and couples linearly to the matter field.

At the mean-field level, the partition function is dominated by the field configurations $\phi_0(\mathbf{x})$ which have the highest weight,

$$\left.\frac{\delta F}{\delta \phi}\right|_{\phi_0(\mathbf{x})} = 0. \tag{60}$$

This is the Euler-Lagrange equation and reads as

$$\frac{\partial f}{\partial \phi} - \boldsymbol{\nabla} \cdot \frac{\partial f}{\partial(\boldsymbol{\nabla}\phi)} = 0. \tag{61}$$

The Ising model belongs to the $\phi^4$ universality class, for which $u_3 = 0$ and $u_4 > 0$, hence there is no need for $u_6$. This is the standard Ginzburg-Landau-Wilson model. Percolation corresponds to $u_3 > 0$ and the expansion is stopped there. The tricritical point which marks the singular behaviour at the end of a line of first-order phase transitions, as e.g. in the Blume-Capel model for a specific choice of parameters, is described by $u_3 = u_4 = 0$, $u_6 > 0$. Note that the gradient term does not have its own coefficient. This is because it is usually absorbed by a rescaling of all other coefficients. This makes this term dimensionless once integrated over space and thus, calling $x_\phi$ the matter-field dimension, one has $2(x_\phi + 1) = d$ which fixes

$$x_\phi = \frac{d}{2} - 1. \tag{62}$$

Compare this to Eq.(49), which is nothing more than power counting, and we see that there is no $\eta$ exponent in mean-field theory. This is why it is termed the *anomalous dimension* — a non-zero value is a measure of deviation from the simplest theory.

Let us consider then a generic model:

$$f(\phi, \boldsymbol{\nabla}\phi) = \tfrac{1}{2}r\phi^2(\mathbf{x}) + \tfrac{1}{n}u_n\phi^n(\mathbf{x}) - h\phi(\mathbf{x}) + \tfrac{1}{2}|\boldsymbol{\nabla}\phi|^2 \tag{63}$$

with $n = 3, 4, 6$ respectively corresponding to percolation, $O(N)$ models (including SAW or polymers) and tricriticality. In an infinite homogeneous system, the gradient term cancels and equation (60) delivers the equation of state

$$\phi_0(r + u_n\phi_0^{n-2}) = h. \tag{64}$$

In the absence of a magnetic field, $h = 0$, the order parameter $m_\infty = \phi_0$ discriminates between the two phases,

$$T < T_c, \qquad m_\infty(t) = (-r/u_n)^{\frac{1}{n-2}}, \tag{65}$$

$$T > T_c, \qquad m_\infty(t) = 0. \tag{66}$$

The high-temperature region where the order parameter vanishes is referred to the "symmetric phase", while the low-temperature phase is called the "symmetry broken phase". This terminology refers to the study of broken symmetries which occur when cooling down a system from its disordered phase at high temperature. In equation (65), one can read the critical exponent of the order parameter. Retaining the terminology of magnetic systems we use the symbol $\beta$,

$$\beta_{\text{MFT}} = \frac{1}{n-2}. \tag{67}$$

The corresponding amplitude is $B^- = (u_n)^{-1/(n-2)}$. At the critical temperature, $r = 0$, the same order parameter has a magnetic field dependence which we extract from (64)

$$T = T_c, \qquad m_\infty(h) = (|h|/u_n)^{\frac{1}{n-1}}. \tag{68}$$

The exponent $\delta$ follows from equation (26),

$$\delta_{\text{MFT}} = n - 1 \tag{69}$$

together with the amplitude $D_c = (u_n)^{-1/(n-1)}$. The susceptibility requires the calculation of the second derivative of (64) wrt $h$, leading to the equation

$$\chi_\infty = [r + (n-1)u_n\phi_0^{n-2}]^{-1} \tag{70}$$

from which one gets two expressions, depending on the values of the order parameter (65) and (66) in each of the two phases:

$$T < T_c, \qquad \chi_\infty(t) = [(n-2)(-r)]^{-1}, \tag{71}$$

$$T > T_c, \qquad \chi_\infty(t) = r^{-1}. \tag{72}$$

Both expressions lead to the same exponent $\gamma$, as anticipated in the definition in equation (27),

$$\gamma_{\text{MFT}} = 1. \tag{73}$$

The associated amplitudes are $\Gamma^- = (n-2)^{-1}$ and $\Gamma^+ = 1$.

The specific heat exponent from equation (25) requires the second derivative of the free energy wrt $t$. The homogeneous system has a free energy given by inserting the equilibrium order parameter (65) and (66) in the expansion (63), leading to

$$T < T_c, \qquad f_\infty = \left(\frac{1}{n} - \frac{1}{2}\right) u_n^{\frac{2}{2-n}} (-r)^{\frac{n}{n-2}}, \tag{74}$$

$$T > T_c, \qquad f_\infty = 0. \tag{75}$$

The specific heat follows, vanishing above $T_c$,

$$T < T_c, \qquad c_\infty = \frac{1}{2-n} u_n^{\frac{2}{2-n}} (-r)^{\frac{4-n}{n-2}}, \tag{76}$$

$$T > T_c, \qquad c_\infty = 0, \tag{77}$$

and a jump appears at the transition, with an exponent associated with the low temperature regime (this is specific to the mean-field solution),

$$\alpha_{\text{MFT}} = \frac{n-4}{n-2}. \tag{78}$$

The amplitude in this regime is $(A^-/\alpha_{\text{MFT}}) = (u_n)^{-2/(n-2)}/(2-n)$.

For the correlations, one has to go back to equation (63) and keep the gradient term. The Euler-Lagrange equation now leads to the differential equation

$$r\phi(\mathbf{x}) - u_n\phi^{n-1}(\mathbf{x}) - \boldsymbol{\nabla}^2\phi(\mathbf{x}) = h. \tag{79}$$

The space dependence of the correlation function corresponds to the order parameter profile when a localized magnetic field $h_0\delta(\mathbf{x})$ is applied at the origin. At criticality, $r = h = 0$ and $\phi(\mathbf{x})$ can be considered small enough to neglect the non-linear term. Outside the origin this leads to a Laplace equation, independently of the value of $n$ at this approximation

$$\boldsymbol{\nabla}^2\phi(\mathbf{x}) = \frac{1}{|\mathbf{x}|^{d-1}}\frac{d}{d|\mathbf{x}|}\left(|\mathbf{x}|^{d-1}\frac{d\phi(\mathbf{x})}{d|\mathbf{x}|}\right) = 0, \tag{80}$$

where isotropy is assumed. The solution is of the form

$$g(\mathbf{x}) \sim \frac{1}{|\mathbf{x}|^{d-2}} \tag{81}$$

and is consistent with the value of the correlation function critical exponent

$$\eta_{\text{MFT}} = 0. \tag{82}$$

This is the standard result of Ornstein-Zernicke theory. This exponent does not depend on $n$ as we anticipated. Outside the critical temperature, (say above $T_c$, since we are still neglecting the $\phi^{n-1}$ term in (79)) the equation to solve is

$$r\phi(\mathbf{x}) - \boldsymbol{\nabla}^2\phi(\mathbf{x}) = h. \tag{83}$$

The approximation for the term in $\phi^{n-1}(\mathbf{x})$ can be questioned, but it is not an essential one in the present context, since we are now reasoning on dimensional arguments. Clearly, one has to introduce length scales, and the only length scale in the thermodynamic limit is the correlation length. We therefore identity the correlation lengths $\xi(t, h = 0)$ and $\xi(t = 0, h)$ as

$$\xi(t, h = 0) \sim (1/r)^{1/2}, \qquad \xi(t = 0, h) \sim (\phi/h)^{1/2}. \tag{84}$$

With $\phi \sim h^{1/\delta}$, both exponents of the correlation length follow:

$$\nu_{\text{MFT}} = 1/2, \qquad \nu_{\text{c MFT}} = \frac{\delta-1}{2\delta} = \frac{n-2}{2(n-1)}. \tag{85}$$

We assume the same temperature dependence for the correlation length below $T_c$ (this can be made more rigorous, see e.g. Ref. [27]). We emphasize that $\nu$ does not depend on $n$.

As special cases, the set of mean-field exponents for the Ising model, percolation and tricriticality universality classes are collected in Table 2.

Lattice animals correspond to another universality class and deserve a few words here, since they fit in the general picture that we are presenting. These are the connected clusters that we can form on a lattice. Like polymers, their sizes and shapes obey specific scaling forms. It happens that they are described by a Landau expansion with a $\phi^3$ theory and only one external field, say a temperature-like field, coupled to the linear power of the matter field $\phi$. We therefore consider

$$f(\phi, \boldsymbol{\nabla}\phi) = r\phi(\mathbf{x}) + \tfrac{1}{n}w_n\phi^n(\mathbf{x}) + \tfrac{1}{2}|\boldsymbol{\nabla}\phi|^2 \tag{86}$$

with $n = 3$ for lattice animals. If we proceed along the same lines as in the previous calculations, we get the order parameter $\phi_0 = (-r/w_n)^{1/(n-1)}$ below $T_c$, hence $\beta_{\mathrm{MFT}} = \frac{1}{n-1}$, or $1/2$ for lattice animals. The susceptibility varies like $\chi \sim 1/\phi^{n-2}$ and delivers $\gamma_{\mathrm{MFT}} = \frac{n-2}{n-1}$ and the specific heat has the same exponent, $\alpha_{\mathrm{MFT}} = \frac{n-2}{n-1}$. The exponent $\eta$ is not modified wrt the theory of Eq. (63), $\eta_{\mathrm{MFT}} = 0$, and dimensional arguments for the correlation length lead to $\xi \sim (\phi/r)^{1/2}$, hence $\nu_{\mathrm{MFT}} = \frac{n-2}{2(n-1)}$. In the absence of another external field in the model, the exponent $\delta$ can be obtained by scaling relations and one gets $\delta_{\mathrm{MFT}} = (\beta + \gamma)/\beta = n - 1$. Collecting all exponents (for $n = 3$), one has the last row in Table 2.

| Model | $\phi^n$ | $\alpha_{\mathrm{MFT}}$ | $\beta_{\mathrm{MFT}}$ | $\gamma_{\mathrm{MFT}}$ | $\delta_{\mathrm{MFT}}$ | $\nu_{\mathrm{MFT}}$ | $\eta_{\mathrm{MFT}}$ | $d_{\mathrm{uc}}$ |
|---|---|---|---|---|---|---|---|---|
| Magnets, SAW | $\phi^4$ | $0$ | $\frac{1}{2}$ | $1$ | $3$ | $\frac{1}{2}$ | $0$ | $4$ |
| Percolation | $\phi^3$ | $-1$ | $1$ | $1$ | $2$ | $\frac{1}{2}$ | $0$ | $6$ |
| Tricriticality | $\phi^6$ | $\frac{1}{2}$ | $\frac{1}{4}$ | $1$ | $5$ | $\frac{1}{2}$ | $0$ | $3$ |
| Lattice animals | $\phi + \phi^3$ | $\frac{1}{2}$ | $\frac{1}{2}$ | $\frac{1}{2}$ | $2$ | $\frac{1}{4}$ | $0$ | $8$ |

Table 2: Mean-field exponents for the Ising model, percolation, tricriticality and lattice animals universality classes.

We said that Landau theory provides a quantitatively valid description of phase transitions when order parameter fluctuations can be neglected. There is a self-consistent criterion which shows that mean-field exponents indeed lead to neglect of fluctuations above a certain space dimension. The fluctuations are essentially measured by the susceptibility, which is the space integral of the order parameter correlation function,

$$\chi \simeq \int d^d x\, g(\mathbf{x}) \sim |t|^{-\gamma_{\mathrm{MFT}}}. \tag{87}$$

This has to be compared to the square of the magnetization inside the correlation volume at the same temperature,

$$\xi^d m_\infty^2 \sim |t|^{-d\nu_{\mathrm{MFT}} + 2\beta_{\mathrm{MFT}}}. \tag{88}$$

If the first expression is dominated by the second, then order parameter fluctuations are weakened. This happens with mean-field values of the exponents when

$$d \geq \frac{2\beta_{\mathrm{MFT}} + \gamma_{\mathrm{MFT}}}{\nu_{\mathrm{MFT}}}. \tag{89}$$

Collecting the values of the mean-field exponents in equations (67), (73) and (85), we get that mean-field theory is valid if

$$d \geq d_{\mathrm{uc}} = \frac{2n}{n-2}. \tag{90}$$

This is known as the Ginzburg criterion and we refer to $d_{\mathrm{uc}}$ as the upper critical dimension. The values of $d_{\mathrm{uc}}$ are given in Table 2.

## 4 The Gaussian fixed point and its apparent failure to describe critical phenomena above $d_{\mathrm{uc}}$

Let us come back to the free energy density (59) or (63) from which we can deduce the matter field scaling dimension (62). Each term has the dimension of a density (per unit volume), and from these the scaling dimensions of the external fields $t, h, u$ follow (from now on, we shorten the notation to $u$ for $u_n$ and $y_u$ for $y_{u_n}$),

$$y_t + 2x_\phi = d, \quad y_t = 2, \tag{91}$$

$$y_h + x_\phi = d, \quad y_h = \frac{d}{2} + 1, \tag{92}$$

$$y_u + nx_\phi = d, \quad y_u = \frac{d}{2}(2-n) + n. \tag{93}$$

These scaling dimensions, or RG eigenvalues, control the renormalization flow of the three parameters:

$$t' = b^{y_t} t, \tag{94}$$

$$h' = b^{y_h} h, \tag{95}$$

$$u' = b^{y_u} u. \tag{96}$$

There is a trivial fixed point at $t = h = u = 0$. It is called the Gaussian Fixed Point (GFP) because there the partition function (57) becomes a Gaussian integral,

$$Z = \int D\phi \, e^{-\int d^d x \, \frac{1}{2} |\nabla \phi|^2}. \tag{97}$$

With the additional scaling field $u$, the homogeneous form (34) becomes

$$f_\infty^{\mathrm{sing}}(t,h) = b^{-d} f_\infty^{\mathrm{sing}}(b^{y_t} t, b^{y_h} h, b^{y_u} u). \tag{98}$$

The simplicity of this scaling form is deceptive as it hides subtle phenomena near the critical point as we shall see. The counterpart forms for the correlation function and correlation length are

$$g_\infty^{\mathrm{sing}}(\mathbf{x}, t, h) = b^{-2x_\phi} g_\infty^{\mathrm{sing}}(b^{-1}\mathbf{x}, b^{y_t} t, b^{y_h} h, b^{y_u} u) \tag{99}$$

and

$$\xi_\infty^{\mathrm{sing}}(t,h) = b \xi_\infty^{\mathrm{sing}}(b^{y_t} t, b^{y_h} h, b^{y_u} u), \tag{100}$$

respectively. The temperature and the magnetic field have positive RG eigenvalues in Eqs.(91) and (91). We say that these are relevant fields. They tell us that under a rescaling by a factor $b > 0$, the relevant fields grow as

$$t' = b^2 t, \quad \text{and} \quad h' = b^{\frac{d}{2}+1} h. \tag{101}$$

Starting from outside the critical point, with either $t \neq 0$ or $h \neq 0$ (or both), renormalization brings the system further and further away from criticality ($t = h = u = 0$). The scaling field $u$ in Eq.(93) also has a positive RG eigenvalue below a certain value

$$d_{\text{uc}} = \frac{2n}{n-2} \tag{102}$$

of the space dimension, meaning that $u$ is relevant there. This is, of course, exactly the value of $d_{\text{uc}}$ in Eq.(90). However, $y_u < 0$ above $d_{\text{uc}}$ and here $u$ is said to be irrelevant. In this case, even if one starts from a non zero initial value of $u$, successive rescalings drive it to zero and leave the system at criticality. The negativity of $y_u$ or irrelevance of $u$ there is another reason why Landau theory and mean-field exponents provide a correct description of critical properties above $d_{\text{uc}}$. The field $u$ in this situation does not determine the universal quantities which maintain the values of the GFP. The border line between the two regimes, precisely at $d_{\text{uc}}$, is the marginal situation where critical singularities are usually accompanied by multiplicative logarithmic divergences.

The question now arises how the above *correct* mean-field exponents (Table 2) compare with the predictions drawn from RG at the Gaussian fixed point. Using the RG eigenvalues (91) and (92), and the scaling dimension (62), inserted in equations (43)–(49), one gets the exponents listed in Table 3 below. The scaling dimensions (91) and (92) take the same values, irrespective of the value of $n$ in the free energy density expansion, i.e. they are the same for all three universality classes — the Ising model, percolation and tricriticality — above their respective upper critical dimensions. The $n$ dependency is carried only by Eq.(93) which, although the field $u$ does not determine Gaussian critical exponents, we also insert in the table.

| $y_t = 2$ | $y_h = \frac{d}{2} + 1$ | $y_u = d(1 - \frac{n}{2}) + n$ |
|---|---|---|
| $\alpha_{\text{G}} = 2 - \frac{d}{2}$ | $\beta_{\text{G}} = \frac{d-2}{4}$ | $\delta_{\text{G}} = \frac{d+2}{d-2}$ |
| $\gamma_{\text{G}} = 1$ | $\nu_{\text{G}} = \frac{1}{2}$ | $\eta_{\text{G}} = 0$ |

Table 3: RG eigenvalues and critical exponents at the Gaussian fixed point. From Eq.(102), $y_u = n(d_{\text{uc}} - d)/d_{\text{uc}} < 0$ if $d > d_{\text{uc}}$ so, in contrast to $y_t$ and $y_h$, is irrelevant there.

The mismatch between Table 2 and Table 3 for some of the exponents is obvious! While the third row of Gaussian exponents in Table 3 do indeed coincide with mean-field exponents, the agreement is broken by the second row. This is clearly a major "flaw in the original [RG] argument" for the RG, since, as we have argued earlier, mean-field exponents are correct above $d_{\text{uc}}$, where $u$ is irrelevant. This is the "failure" that Fisher was referring to in the second quote of Section 2 above.

In trying to understand the origin of the failure of the GFP above $d_{\text{uc}}$, we notice that the (wrong) exponents $\alpha$, $\beta$ and $\delta$ all come from derivatives of the free energy, while the (right) $\gamma$, $\nu$ and $\eta$ are associated with the correlations. We will refer to the first set of quantities as belonging to the "free energy sector" and the second group as belonging to the "correlation sector". The susceptibility is special in the sense that it is at the same time associated with a free energy derivative and the correlation function integral. Still, an explanation of the discrepancy is needed for the free energy sector. (We will see that it is also needed for

the apparently unproblematic correlation sector too where extremely subtle incompatibilities hide.) Incidentally, we can also observe that all exponents, including the obviously "deviant" ones coincide with the mean-field counterparts precisely at the respective values of $d_{\text{uc}}$ for the three universality classes discussed here. In addition, the GFP value for $\nu_c$ is $\nu_{c\,\text{G}} = 2/(d+2)$ from inserting Eq.(92) for $y_h$ into Eq.(48), and this does not agree with the mean-field value in Eq.(85) except, again, when $d = d_{\text{uc}}$. This is the sign that something is wrong also in the correlation sector!

## 5  The Dangerous Irrelevant Variable scenario

### 5.1  Fisher's breakthrough

In 1983,[3] revisiting the question of discordance between the Gaussian fixed point and mean-field theory, Michael Fisher [2] made a very smart observation. Although one would expect the GFP to deliver the correct predictions above $d_{\text{uc}}$ (where the exponents should not depend on $u$), Fisher noticed that for the three quantities in the "free energy sector" that lead to the exponents $\alpha$, $\beta$ and $\delta$, the limit $u \to 0$ in the expressions obtained in Landau theory is problematic. This comes from the amplitudes in equations (65), (68) and (76) which are singular when $u \to 0$,

$$B^- = u^{-\frac{1}{n-2}}, \tag{103}$$

$$D_c = u^{-\frac{1}{n-1}}, \tag{104}$$

$$\frac{A^-}{\alpha_{\text{MFT}}} = \frac{1}{2-n} u^{-\frac{2}{n-2}}. \tag{105}$$

The irrelevant field $u$ is therefore *dangerous* and we speak about the role of the *dangerous irrelevant variable* (DIV). Not addressing these is the "failure to recognize and allow for possible singular behaviour of the scaling function" that Fisher was referring to in the quote of Sec.2. The amplitudes of the quantities in the "correlation sector", on the other hand, do not depend on $u$ and do not face the same difficulty (with the notable exception of $\xi_c$, but the $h-$dependence of the correlation length at $T_c$ is rarely discussed, so we also leave this quantity aside). Another argument given by Fisher and Privman [1] is that a strictly positive value of $u$ is required to ensure the stability of the free energy.

Here we introduce the following notation to compactify the exponents appearing in (103)–(105):

$$m_\infty(t < 0, h = 0, u) \sim |t|^\beta u^{-\kappa}, \tag{106}$$

$$m_\infty(t = 0, h, u) \sim |h|^{1/\delta} u^{-\lambda}, \tag{107}$$

$$c_\infty(t, h = 0, u) \sim |t|^{-\alpha} u^{-\mu}, \tag{108}$$

where in $\phi^n$ Landau theory, these parameters take the values

$$\kappa = \frac{1}{n-2}, \quad \lambda = \frac{1}{n-1}, \quad \mu = \frac{2}{n-2}. \tag{109}$$

These are collected in Table 4 for the universality classes under consideration.

---

[3]Although the proceedings were published in 1983, the lecture itself was given in 1982. The idea probably germinated in Fisher's mind a lot earlier as another lecture dating from 1973 is often cited in this context [28].

| Model | $\phi^n$ | $\kappa$ | $\lambda$ | $\mu$ |
|-------|----------|----------|-----------|-------|
| Magnets, SAW | $\phi^4$ | $\frac{1}{2}$ | $\frac{1}{3}$ | $1$ |
| Percolation | $\phi^3$ | $1$ | $\frac{1}{2}$ | $2$ |
| Tricriticality | $\phi^6$ | $\frac{1}{4}$ | $\frac{1}{5}$ | $\frac{1}{2}$ |

Table 4: Exponents of the dangerous irrelevant variable in Landau theory.

To accommodate these observations, equations (39) and (42) have to be modified, taking into account the dependence on the DIV $u$, and its dangerous limit of $u \to 0$. E.g., to draw the singularity in the scaling function for the magnetisation to the fore we express it as

$$m_\infty(x,0,z) \overset{u\to 0}{=} z^{-\kappa}\mathscr{M}^-(x,0). \tag{110}$$

The scaling hypothesis for the magnetization and specific heat are now modified and must obey

$$m_\infty(t,0,u) \overset{u\to 0}{=} b^{-d+y_h-\kappa y_u}u^{-\kappa}\mathscr{M}^-(b^{y_t}t,0), \tag{111}$$

$$m_\infty(0,h,u) \overset{u\to 0}{=} b^{-d+y_h-\lambda y_u}u^{-\lambda}\mathscr{M}_c(0,b^{y_h}h), \tag{112}$$

$$c_\infty(t,0,u) \overset{u\to 0}{=} b^{-d+2y_t-\mu y_u}u^{-\mu}\mathscr{C}^\pm(b^{y_t}t,0). \tag{113}$$

On the other hand, nothing has to be modified for the "correlation sector" (susceptibility, correlation function, correlation length). Fixing the scaling factor $b$ to the appropriate value, $|t|^{-1/y_t}$ or $|h|^{-1/y_h}$ in (111)–(113), then leads, to

$$\beta_{\mathrm{MFT}} = \frac{d-y_h}{y_t} + \frac{\kappa y_u}{y_t} = \frac{1}{n-2}, \tag{114}$$

$$\frac{1}{\delta_{\mathrm{MFT}}} = \frac{d-y_h}{y_h} + \frac{\lambda y_u}{y_h} = \frac{1}{n-1}, \tag{115}$$

$$\alpha_{\mathrm{MFT}} = \frac{2y_t-d}{y_t} - \frac{\mu y_u}{y_t} = \frac{n-4}{n-2} \tag{116}$$

which completely repairs the free energy sector above $d_{\mathrm{uc}}$. The values of $\kappa$, $\lambda$ and $\mu$ in Eq.(109) are precisely those that match Eqs.(67), (69) and (78), which are gathered here for convenience. The amplitudes also should be consistent and we find for example for the magnetization approaching the critical temperature from below

$$B^- = \mathscr{M}^-(0^-)u^{-\kappa}. \tag{117}$$

We have now reached a point where RG appears to be fully successful in its treatment of critical properties above the upper critical dimension in the thermodynamic limit at least. To refer again to the quote in Sec. 2, "the renormalization group framework has been preserved intact" — or so it seems. However, besides for notational purposes in the prelude (Sec. 1), we have so far not touched on finite-size systems. We address finite-size scaling in the next section and encounter another conflict that again raises question about RG. The dichotomy this time is between predictions coming from MFT and analytical results from Brézin et al. [29] as well as numerical results from Binder et al. [30], all dating from the 1980's. We present these in the next section along with partial solutions up to the 1990's.

## 5.2 The problem with finite-size scaling

In the standard theory of finite-size scaling (FSS), if, in the thermodynamic limit, a physical quantity $Q(t, h)$ is described by a critical exponent $\rho$ wrt temperature, say, i.e.,

$$Q_\infty(t, 0) \sim |t|^\rho, \tag{118}$$

one usually takes that its finite-size counterpart $Q_L(t, 0)$ at a given temperature $t$ is controlled by the ratio of the finite size $L$ to the typical length scale which governs criticality, the correlation length $\xi_\infty(t)$ at the same temperature,

$$Q_L(t, 0) = Q_\infty(t, 0) f_Q(L/\xi_\infty(t)). \tag{119}$$

The finite system cannot deliver any singularity because the partition function is a finite sum of exponentials and can only display regular behaviour. Therefore we demand that the function $f_Q(x) \sim x^\omega$ corrects the singularity in $Q_\infty$. This in turn implies that $\omega = -\rho/\nu$, and we obtain the FSS behaviour $Q_L(t, 0) = A_Q(t) L^{-\rho/\nu}$. To get rid of the temperature dependent prefactor, one usually fixes $t = 0$ to sit at the critical point and obtain

$$Q_L(t = 0, 0) \sim L^{-\rho/\nu}. \tag{120}$$

The argument is also encoded in the scaling hypothesis and is probably more convincing there. Using the case of the susceptibility for example, the choice $b = L$ and $h = 0$ in equation (40) leads to

$$\chi_L(t, 0) = L^{\frac{\gamma}{\nu}} \mathscr{X}(L^2 t). \tag{121}$$

At the critical point $t = 0$ this gives $\chi_L(t = 0) \sim L^{\frac{\gamma}{\nu}}$, but usually FSS is performed at the pseudo-critical point $t_L$ instead. This is easier to determine in finite-size numerical simulations than the true critical temperature which requires extrapolation to the thermodynamic limit. The pseudo-critical temperature $T_L$ can be defined by, e.g., the value of the temperature for which the finite-size susceptibility (or any other diverging quantity) reaches its maximum, $\chi_L(T_L) = \text{Max}_T \chi_L(T)$, a quantity often denoted in the literature as $\chi_{\max}(L)$, but that we will call here $\chi_L(t_L, 0)$ with $t_L = T_L - T_c$, the second argument being the magnetic field, as usual. There, an expansion of (121) leads to $\chi_L(t_L, 0) \simeq L^{\frac{\gamma}{\nu}} \mathscr{X}(t = 0) + \dots$ and thus, up to corrections to scaling, to the same leading FSS behaviour as at $T_c$.

Inserting MFT exponents in equation (120), one thus expects the FSS behaviour

$$c_L(t = 0, 0) \sim L^{2\frac{n-4}{n-2}}, \tag{122}$$

$$\chi_L(t = 0, 0) \sim L^2, \tag{123}$$

$$m_L(t = 0, 0) \sim L^{-\frac{2}{n-2}}, \tag{124}$$

$$\xi_L(t = 0, 0) \sim L. \tag{125}$$

We call this *Landau FSS* because the exponents which appear in powers of $L$ are all ratios of exponents from Landau theory.

These predictions, however, fail. The first theoretical analysis of finite-size correlation length for the $\phi^4$ model above $d_{\text{uc}} = 4$ dimensions was reported in the early 1980's by Brézin [29], who obtained

$$\xi_L(t = 0, 0) \sim L^{d/4} \tag{126}$$

which contradicts Landau scaling (125). Brézin had considered hypercubic systems with periodic boundary conditions. The same author, with Zinn-Justin, then (1985) produced a more complete study of FSS in phase transitions in the $\phi^4$ model [31] and reported for example the susceptibility behaviour, above $d_{\rm uc}$,

$$\chi_L(t=0,0) \sim L^{d/2}, \tag{127}$$

with the comment that "usual FSS does not hold". These authors conclude their paper with

> *It seems clear that in spite of the extensive literature on the subject, there is still a lot to say about finite-size effects.*

In early numerical studies (1985), Binder reported results for the finite-size susceptibility of the Ising model in 5 dimensions with PBC's [30] (see also [32]). In particular, at the pseudo-critical point he obtained

$$\chi_L(t_L,0) \sim L^{5/2}, \quad \text{and} \quad t_L \sim L^{-5/2}. \tag{128}$$

Binder also led a discussion of the specific heat maximum there, but we believe that due the available sizes being too small ($L \leq 7$), the conclusions were probably not sound (in such early studies the thermal sector was not fully understood). The relation (126) of Brézin was later checked numerically by Jones and Young [33].

Later, other quantities have been calculated via MC simulations by many people (see the review [34] and references therein), and, if we quote only the results for the $5d$ Ising model with PBC's, we collect

| | pseudo-critical point | critical point | |
|---|:---:|:---:|:---:|
| correlation length | $\xi_L(t_L,0) \sim L^{5/4}$, | $\xi_L(t=0,0) \sim L^{5/4}$, | (129) |
| susceptibility | $\chi_L(t_L,0) \sim L^{5/2}$, | $\chi_L(t=0,0) \sim L^{5/2}$, | (130) |
| magnetization | $m_L(t_L,0) \sim L^{-5/4}$, | $m_L(t=0,0) \sim L^{-5/4}$, | (131) |
| pseudo-critical temperature | $t_L \sim L^{-5/2}$, | | (132) |
| rounding of the critical point | $\Delta\beta_{\chi_{\rm max}/2} \sim L^{-5/2}$, | | (133) |
| pseudo-critical magnetic field | $\|h_L\| \sim L^{-15/4}$, | | (134) |
| Lee-Yang zero | $h_L^{\rm LY}(t_L) \sim L^{-15/4}$, | $h_L^{\rm LY}(t=0) \sim L^{-15/4}$, | (135) |
| Fisher zero | | $t_L^{\rm F}(h=0) \sim L^{-5/2}$. | (136) |

We have discussed the first four rows and the other quantities require some explanation.

The rounding $\Delta\beta_{\chi_{\rm max}/2}$ is the width of the temperature window as measured at half the susceptibility height. The quantity denoted as $|h_L|$ is the shift of magnetic field at the critical temperature, i.e. the finite value of the magnetic field ($\pm|h_L|$) at which the susceptibility peaks at $t=0$. (This does not occur in zero magnetic field). Finally, $h^{\rm LY}$ is the first Lee-Yang zero and $t^{\rm F}$ the first Fisher zero, to be discussed in Section 6.

There is clearly a disagreement above $d_{\rm uc}$ with Landau FSS, and it does not seem to be solved by Fisher's DIV mechanism. Even a desperate attempt to invoke the (wrong) GFP values for the critical exponents of Table 1 fails to rescue the situation; while they deliver different FSS in the free-energy sector (namely, $c_\infty \sim L^{\alpha_G/\nu_G} = L^{4-d}$ and $m_\infty \sim L^{-\beta_G/\nu_G} = L^{-(d-2)/2}$), they deliver the same as Landau FSS for $\chi$ and $\xi$. And, as we have seen, these are $n$-independent and not in agreement with exact or numerical results for the Ising case of $n=4$. This calls for further developments and this is the motivation of Binder, Nauenberg, Privman and Young's (BNPY) approach [35].

## 5.3 Dangerous irrelevancy for the free energy and the thermodynamic length

In 1985, Binder, Nauenberg, Privman and Young suggested an extension of the DIV mechanism [35], but we believe that it was not fully developed. It results in a theory compatible with Fisher's and, indeed, Landau MFT, and involves an additional hypothesis that at least partially solves the problem of FSS. We first present this approach and the way in which finite-size effects are understood in BNPY's theory and then we expose what we believe are still weak points calling for further extensions of Fisher's DIV concept.

We have seen with Fisher's analysis that the homogeneity assumption has to be modified to take into account the existence of the DIV $u$. BNPY suggested to build on Fisher's suggestion to reconsider, e.g., the magnetization and the specific heat, assuming that not only the prefactors, but also the rescaled arguments of the free energy might be affected. The modification propagates to the dimension of prefactors in the free energy derivatives. They made the following hypothesis in the dangerous limit:

$$f_\infty^{\text{sing}}(x, y, z) \stackrel{z \to 0}{=} z^{p_1} f_\infty^{\text{sing}}(xz^{p_2}, yz^{p_3}, 0) \tag{137}$$

which can be rewritten in a more convenient form [cf Eq.(34)]:

$$f_\infty^{\text{sing}}(t, h, u) \stackrel{u \to 0}{=} b^{-d + p_1 y_u} u^{p_1} \mathscr{F}^\pm (b^{y_t + p_2 y_u} t u^{p_2}, b^{y_h + p_3 y_u} h u^{p_3}). \tag{138}$$

Here $p_1$, $p_2$ and $p_3$ are constants which have to be determined by further considerations. BNPY also introduced the notations

$$d^* = d - p_1 y_u, \quad y_t^* = y_t + p_2 y_u, \quad y_h^* = y_h + p_3 y_u, \tag{139}$$

and also assumed that the correlation length could obey a similar homogeneity law,

$$\xi_\infty(t, h, u) \stackrel{u \to 0}{=} b^{1 + q_1 y_u} u^{q_1} \Xi (b^{y_t + q_2 y_u} t u^{q_2}, b^{y_h + q_3 y_u} h u^{q_3}), \tag{140}$$

with the three other parameters, $q_1$, $q_2$ and $q_3$, at that point unknown.

Binder and his coauthors then developed an argumentation to support the values (they presented their results for $n = 4$ but we generalise them here

$$p_1 = 0, \tag{141}$$

$$q_1 = 0. \tag{142}$$

The first result follows from the assumption $d^* = d$, and is underpinned by a discussion on the zero-field susceptibility for a finite system. The second result is supported as follows:

> *Since the finite-size correlation length $\xi_L$ is bounded by $L$, we require $q_1 y_u < 0$. (...) if one adopts the plausible assumption that for $t = h = 0$, the correlation length increases up to the linear dimensions of the lattice, which implies that $q_1 = 0$.*

We will see later that the value $q_1 = 0$ is not correct.

The values of $p_2$ and $p_3$ are not explicitly written in BNPY's paper, but an immediate consequence of the work reported for the $\phi^4$ case considered there is

$$p_2 = -\tfrac{1}{2}, \quad p_3 = -\tfrac{1}{4}. \quad \text{(These formulae are for } n = 4 \text{ only.)} \tag{143}$$

In particular, BNPY discussed the scaling with the values $y_t^* = d/2$ and $y_h^* = 3d/4$ (for $n = 4$, again) which fix $p_2$ and $p_3$. On the other hand, they did not pursue a discussion of $q_2$ and $q_3$, leaving the option that these parameters might differ from $p_2$ and $p_3$, an opinion that we do not share, as we discuss later.

Although this is not done in the original BNPY paper, we believe it is instructive to explicitly state the complete agreement with Fisher's scenario. Here we do this in the context of the $\phi^n$ model of equation (63). Using the appropriate derivatives of the free energy density wrt $t$, and to $h$, we get the first four "classical" exponents in the BNPY approach

$$\alpha_{\mathrm{MFT}} = \frac{2y_t^* - d}{y_t^*}, \quad \beta_{\mathrm{MFT}} = \frac{d - y_h^*}{y_t^*}, \tag{144}$$

$$\gamma_{\mathrm{MFT}} = \frac{2y_h^* - d}{y_t^*}, \quad \delta_{\mathrm{MFT}} = \frac{y_h^*}{d - y_h^*}. \tag{145}$$

The comparison with the results of Fisher, (116), leads to the following results for the $p_i$'s parameters (Table 5),

$$p_1 = 0, \quad p_2 = -\frac{2\kappa y_t}{d + 2\kappa y_u} = -\frac{2}{n}, \quad p_3 = -\frac{\kappa(2y_h - d)}{d + 2\kappa y_u} = -\frac{1}{n}, \tag{146}$$

having used Eq.(109) for the general $n$ case. Inserting these values in Eq.(139) gives

$$d^* = d, \quad y_t^* = y_t - \frac{2y_u}{n}, \quad y_h^* = y_h - \frac{y_u}{n}, \tag{147}$$

In terms of $d$ and $n$ these scaling dimensions are

$$y_t^* = \frac{d(n-2)}{n}, \quad y_h^* = \frac{d(n-1)}{n}. \tag{148}$$

Inserting $y_t^*$ for $y_t$ and $y_h^*$ for $y_h$ in (43) (44) (45) in (46) delivers the correct Landau MFT critical exponents for $\alpha$, $\beta$, $\delta$ and $\gamma$ in (67), (69), and (73), (78). The remaining main critical exponents $\nu$ and $\eta$ are unaffected because the correlation sector is not touched upon in Eq.(138).

| Model | $\phi^n$ | $p_2$ | $p_3$ | $y_t^*$ | $y_h^*$ |
|---|---|---|---|---|---|
| Magnets, SAW | $\phi^4$ | $-\frac{1}{2}$ | $-\frac{1}{4}$ | $\frac{d}{2}$ | $\frac{3d}{4}$ |
| Percolation | $\phi^3$ | $-\frac{2}{3}$ | $-\frac{1}{3}$ | $\frac{d}{3}$ | $\frac{2d}{3}$ |
| Tricriticality | $\phi^6$ | $-\frac{1}{3}$ | $-\frac{1}{6}$ | $\frac{2d}{3}$ | $\frac{5d}{6}$ |
| General | $\phi^n$ | $-\frac{2}{n}$ | $-\frac{1}{n}$ | $(1 - \frac{2}{n})d$ | $(1 - \frac{2}{n})d$ |

Table 5: Renormalization of the RG eigenvalues by the dangerous irrelevant variable in BNPY theory, extended here to percolation theory and to tricriticality.

Thus far we have formulated Fisher's DIV concept for the free energy itself (in the thermo-dynamic limit). We have still not touched the (seemingly unbroken) correlation sector. What is probably the most important assumption made in BNPY paper, as well as in Ref. [30],

concerns FSS. The authors were aware of the failure above $d_{\text{uc}}$ of the ordinary scenario which works well below $d_{\text{uc}}$ and proposed to repair it via the introduction of a new length scale, the *thermodynamic length* $l(t, h)$ which, instead of the correlation length, would be the relevant scale which controls finite-size effects there. Various reasons in favor of the thermodynamic length were given and in particular the fact that such a length scale, $l \sim |t|^{-1/y_t^*}$, appears in (138) when the first argument is written in the dimensionless form $(L/l)^{y_t^*}$. With this new hypothesis to hand, equation (119) would be replaced, above $d_{\text{uc}}$ by

$$Q_L(t, 0) = Q_\infty(t, 0) f_Q(L/l(t)) \tag{149}$$

leading to the FSS behaviour

$$Q_L(t = 0, 0) \sim L^{-\rho y_t^*} \tag{150}$$

instead of (120), and in the particular case of the susceptibility for the Ising model, to

$$\chi_L(t = 0, 0) \sim L^{\gamma y_t^*} = L^{d/2} \tag{151}$$

where $\gamma = \gamma_{\text{MFT}} = 1$ and $y_t^* = d/2$ are used. This result conforms to the numerical simulations of Binder [30]. These simulations are also consistent with $\gamma = \gamma_{\text{MFT}} = 1$.

At this point the mismatch between RG and analytical/numerical results for the Ising susceptibility appears fixed. We have to stress that in their paper, BNPY did not discuss the FSS behaviour of the correlation length, but their result, $q_1 = 0$, is incompatible with the calculation of Brézin [29] (see also [36]), as we will discuss later in this paper. We believe that the main reason for this wrong result, which was pretty common even up to recent times, is probably due to an erroneous belief concerning the very nature of the correlation length. This length is not a *material* length. This is an abstract length defined as the typical scale of an exponential decay. From this point of view, there is no need to demand physical limitations like the system size. In our opinion, if one considers that a DIV affects the free energy density, possibly modifying the usual $b^{-d}$ prefactor, one can just as well contemplate that it affects the correlation length, possibly changing its standard $b$ prefactor. In fact, BNPY showed that the $b^{-d}$ prefactor in the free-energy case is unchanged. This may influence one's expectation concerning the correlation length, but we believe that the modification of the correlation length prefactor is still as legitimate as the one of the magnetization, or of the susceptibility for example. We address that in Section 9.

## 5.4 Dangerous irrelevancy and finite-size corrections

Erik Luijten and Henk Blöte [37, 38], using a RG analysis, provided a wonderful explanation of Fisher and BNPY's DIV scenarios, which otherwise seem ad-hoc in nature. In his doctoral thesis, a tour de force developed under the direction of Blöte, Luijten made a sound analysis of the 4D and 5D Ising models, including with Long-range Interactions, above their upper critical dimensions [39]. The approach of Luijten and Blöte provides a more complete mechanism, and almost closes the question concerning finite-size-scaling above the upper critical dimension at least for the free energy sector. In their original paper, they considered the $O(N)$ $\phi^4$ model and here we extend the results to the $\phi^n$ model of equation (63). The study is based on RG

equations for the scaling fields which, limited to linear order, are of the form[4]

$$\frac{dr}{d\ln b} = y_t r + pu, \tag{152}$$

$$\frac{du}{d\ln b} = y_u u, \tag{153}$$

$$\frac{dh}{d\ln b} = y_h h. \tag{154}$$

Integration of these equations leads to

$$u' = b^{y_u} u, \tag{155}$$

$$r' = b^{y_t} \left( r - \frac{p}{(y_u - y_t)} u + \frac{p}{(y_u - y_t)} b^{y_u - y_t} u \right), \tag{156}$$

$$h' = b^{y_h} h. \tag{157}$$

The details of the integration with the inclusion of non linear terms on the RHS of (153) and (152) can be found in Refs. [39, 40]. The notation is simplified via the introduction of a constant

$$\tilde{p} = -\frac{p}{(y_u - y_t)} \tag{158}$$

and the identification of the temperature scaling field,

$$t = r + \tilde{p}u \tag{159}$$

leading to the expression

$$r' = b^{y_t}(t - b^{y_u - y_t}\tilde{p}u). \tag{160}$$

Now, we define the second moment of the order parameter

$$\langle \phi^2 \rangle = \frac{\int d\phi \, \phi^2 \, e^{-F[\phi]}}{\int d\phi \, e^{-F[\phi]}} \tag{161}$$

where, for homogeneous systems, $F[\phi] = L^d(\frac{1}{2}r\phi^2 + \frac{1}{n}u\phi^n - h\phi)$. A change of variable

$$\phi = u^{-\frac{1}{n}}\varphi \tag{162}$$

absorbs the DIV and leads to $F[\varphi] = L^d(\frac{1}{2}r^*\varphi^2 + \frac{1}{n}\varphi^n - h^*\varphi)$ where

$$r^* = ru^{-\frac{2}{n}} \quad \text{and} \quad h^* = hu^{-\frac{1}{n}} \tag{163}$$

and is such that

$$\langle \phi^2 \rangle = u^{-\frac{2}{n}} \langle \varphi^2 \rangle. \tag{164}$$

Equations (156) and (157) are then transposed to a rescaling of $r^*$ and $h^*$,

$$\begin{aligned} r^{*\prime} &= r'u'^{-\frac{2}{n}} \\ &= b^{y_t - \frac{2}{n}y_u}(t - b^{y_u - y_t}\tilde{p}u)u^{-\frac{2}{n}}, \end{aligned} \tag{165}$$

$$\begin{aligned} h^{*\prime} &= h'u'^{-\frac{1}{n}} \\ &= b^{y_h - \frac{1}{n}y_u}hu^{-\frac{1}{n}}. \end{aligned} \tag{166}$$

---

[4]In Ref. [39], the parameter $p$ is denoted $3ac$, and since Luijten and Blöte consider the long-range interaction model (discussed later), $y_t = \sigma$ and $y_u = 2\sigma - d = \epsilon$.

This expression shows how the DIV contaminates the temperature and magnetic field, and modifies accordingly the homogeneity assumption for the singular part of the free energy density, including also the system size as an additional scaling field. It reads now as

$$f_L(t, h, u) = L^{-d} \mathscr{F}^{\pm}[L^{y_t^*}(tu^{-2/n} - \tilde{p}u^{(n-2)/n}L^{y_u-y_t}), L^{y_h^*}hu^{-1/n}] \tag{167}$$

with $y_t^*$ and $y_t^*$ defined in Eq.(147.) In (167), $L$ is the finite linear scale of the sample (say the length of the edge of a hypercube), and the free energy density is no longer singular.

In our opinion, the analytic inclusion of corrections to scaling in equation (167) is really a central result. The two-term structure was already proposed in BNPY for FBC's but not for PBC's. While they proposed that the free energy scales as Eq.(137) (with $b = L$) for PBC's, "for other boundary conditions, where the system has a surface, it is probably necessary to use both $tL^{y_y^*}$ and $tL^{1/\nu}$ for a complete asymptotic description" [35]. The second term in Eq.(169) when $n = 4$ corresponds to BNPY's proposal and extends it to other boundary conditions, including PBC's.

Eq.(167) contains essentially FSS as a natural consequence. Luijten and Blöte for example demonstrate the size dependence of the shift of the pseudo-critical temperature in the following way. If we set the first argument of (167) scaled by $L$ as

$$X = L^{y_t^*}(tu^{-2/n} - \tilde{p}u^{(n-2)/n}L^{y_u-y_t}), \tag{168}$$

we can denote by $X_0$ the value taken by this variable when a diverging quantity, say the susceptibility, reaches its maximum value wrt the temperature for a finite system. This temperature is, by definition, the pseudo-critical temperature $t_L$, which thus obeys the following scaling

$$t_L = X_0 u^{2/n}L^{-y_t^*} + \tilde{p}uL^{y_u-y_t}. \tag{169}$$

If the first term dominates, which will always occur for some large $L$, because $y_t - y_u \geq y_t^*$ above $d_{\mathrm{uc}}$, we recover, in the case of the Ising model, the FSS of Binder in (128) for $d = 5$.

At this point the homogeneity assumption for the free energy sector has been modified and the size-dependency of the pseudocritical temperature has entered the game. We will later develop the theory of Luijten and Blöte but we present now three additional elements which were studied more recently.

## 6  Zeros of the partition function

We return to the partition function (2) of Sec.1 and, e.g., for the Ising model define it as a sum over the configurations of the microscopic degrees of freedom $\{s_i\}$:

$$Z = \sum_{\{s_i\}} e^{\beta J \sum_{(i,j)} s_i s_j + \beta H \sum_i s_i}. \tag{170}$$

The partition function encodes the thermodynamic properties through its relation to the free energy $F = -\beta^{-1} \ln Z$, and when the latter becomes singular at a phase transition, the former approaches zero. The zeros in the complex magnetic-field plane are called Lee-Yang zeros [41,42] and those in the complex temperature field are called Fisher zeros [43]. Lee-Yang and Fisher zeros are, in a sense, the most fundamental quantities in terms of which most of the thermodynamics quantities can be defined — so much so that they form the basis of what

is referred to as the "fundamental theory of phase transitions" [44]. Any exposition of the fundamentals of RG and scaling theory should incorporate them and that is the aim of this section.

We discuss first the Lee-Yang zeros. For convenience, we keep the Ising model as an example and we define new degrees of freedom $\sigma_i = \frac{1}{2}(1+s_i)$ which take the values 0 and 1. The partition function of a finite system with $N = L^d$ sites is rewritten as a sum over microstates with defined values of the energy $E = -\sum_{(i,j)} s_i s_j$ ($E$ can take positive and negative integer values ranging from $-dN$ and $+dN$) and the magnetization $S = \sum_i s_i = 2\sum_i \sigma_i - N = 2M - N$ (where $M = \sum_i \sigma_i$ takes positive integer values from 0 to $N$). Then, the partition function reads as

$$Z_L(\beta_c) = \sum_{M=0}^{N} \sum_{E=-dN}^{dN} p(E, M) e^{-\beta_c H N} e^{-\beta_c(E+2HM)} = e^{-\beta_c H N} \sum_{M=0}^{N} g_M z^{2M} \quad (171)$$

with the fugacity $z = e^{-\beta_c H}$, and the $g_M$'s are positive numbers (degeneracies of the microstates) which do not depend on $H$. One thus observes that equation (171) is a $N$-th order polynomial in $z^2$ with positive coefficients. As a consequence, $Z_L(\beta_c) = 0$ is an equation which has only complex roots $z^{(k)}$. This is consistent with the fact that, for a finite system, the free energy is analytic at any real value of the magnetic field $H$, hence the partition function has no zero for real $H$. Therefore, if we allow for complex values of the magnetic field, $z = e^{g+ih}$ ($g, h \in \mathbb{R}$), $Z_L(\beta_c)$ has $N$ non-real zeros $z^{(k)}$ in the complex plane. This allows us to factorize

$$Z_L(\beta_c) = A(z) \prod_{k=1}^{N} (z - z_L^{(k)}(\beta_c)), \quad (172)$$

$$z^{(k)} = r^{(k)} e^{i\phi^{(k)}}, \quad \text{with} \quad r^{(k)} = e^{-2\beta_c \Re(H)} = e^{g^{(k)}}, \ \phi^{(k)} = -2\beta_c \Im(H) = h^{(k)} \quad (173)$$

with $A(z)$ a smooth non-vanishing function and $\Im(H_n) \neq 0$. The Lee-Yang theorem states that the zeros lie on the unit circle of the variable $z$, i.e. $z^{(k)} = e^{i\phi^{(k)}}$ or, along the imaginary axis of the variable $H$ [41].

When the system size increases, the order of the polynomial increases, and with it the number of zeros. In the thermodynamic limit, $L \to \infty$, the zeros on the imaginary axis of the variable $g + ih$ become dense. There is a gap between the real magnetic-field axis and a certain point $ih^{\mathrm{LY}}(t)$ for $\beta < \beta_c$. That point is called the Lee-Yang edge. A phase transition occurs in zero magnetic field at $\beta_c$, which means that the gap vanishes when $\beta$ approaches the critical point $\beta = \beta_c$. The vanishing of the gap is controlled by a power law, involving the so-called gap exponent $\Delta$, so that $h^{\mathrm{LY}}(t) \sim |t|^{\Delta}$. The finite-size scaling of the Lee-Yang edge follows from ordinary scaling: the rescaled arguments of the free energy density are $b^{y_t} t$ and $b^{y_h} h$, and, with the choice $b = |t|^{-1/y_t}$, they become 1 and $|t|^{-y_h/y_t} h$, showing that the correct scaling between $h$ and $t$ is indeed

$$h^{\mathrm{LY}}(t) \sim |t|^{\Delta}, \quad \text{where} \quad \Delta = y_h/y_t = \beta\delta = \beta + \gamma. \quad (174)$$

The expression (174) can be written as $h^{\mathrm{LY}}(t) \sim \xi^{-\Delta/\nu}$ which, for a finite system below the upper critical dimension translates into $h_L^{\mathrm{LY}} \sim L^{-\Delta/\nu}$. For finite systems, Itzykson et al [45] showed that zeros approximately scale at the critical temperature, according to their rank $k$, as

$$h^{(k)} \sim (k/L^d)^{\Delta/\nu d}. \quad (175)$$

From equation (172), we get

$$f_L(0, h) = -L^{-d} \sum_k \ln(z - z_L^{(k)}(\beta_c)), \tag{176}$$

$$\chi_L(0, h) = -L^{-d} \sum_k \frac{1}{(z - z_L^{(k)}(\beta_c))^2} \to_{H \to 0} N^{-1} \sum_{k=1}^N (1 - z_L^{(k)}(\beta_c))^{-2} \tag{177}$$

where the last term on the right of (177) is evaluated in zero magnetic field. It follows that

$$\chi_L(t = 0, 0) \simeq \text{const} \times N^{-1} \sum_{k=1}^N (h_L^{(k)}(\beta_c))^{-2} \simeq N^{2\Delta/d\nu - 1} \tag{178}$$

where the sum is dominated by the lowest zeros. This leads to $\chi_L \sim L^{(2\Delta - d\nu)/\nu}$ which is consistent with the ordinary FSS behaviour of the susceptibility, $\chi_L \sim L^{\gamma/\nu}$, when we make use of the scaling relations $\Delta = \beta + \gamma$, $d\nu = 2 - \alpha$ and $\alpha + 2\beta + \gamma = 2$.

Above the upper critical dimension, the Landau exponents of Section 3 suggest

$$\Delta_{\text{MFT}} = \frac{n - 1}{n - 2} \tag{179}$$

for the mean-field gap exponent. This is $\Delta_{\text{MFT}} = \frac{3}{2}$ for the Ising universality class. However, we now know that the scaling $\chi_L \sim L^{\gamma/\nu}$ with MFT exponents is not correct, and above $d_{\text{uc}}$, we have to make appropriate adjustments. FSS now suggests

$$h_L^{(k)} \sim L^{-y_h^*} \tag{180}$$

from where the susceptibility follows as

$$\chi_L(t = 0, 0) \sim L^{2y_h^* - d}. \tag{181}$$

The scaling form (180) for the Lee-Yang edge has been checked for the Ising model in $d = 5$ dimensions in Ref. [34].

The Fisher zeros at $H = 0$ in the complex temperature plane are analyzed in a similar manner. They have been studied in Flores-Sola's doctoral thesis which is openly available at [40]. There, using obvious notations, one expects above the upper critical dimension FSS to be

$$t_L^{(k)} \sim L^{-y_t^*}. \tag{182}$$

This scaling form has been confirmed numerically in the case of the long-range Ising model discussed later. This is consistent with the FSS of the specific heat

$$c_L(t = 0, 0) \sim L^{2y_t^* - d}. \tag{183}$$

# 7 Scaling of the Fourier modes

The Ginzburg-Landau-Wilson action (58), with the Lagrangian density of the $\phi^4$ theory, namely

$$F_{\text{GLW}}[\phi(\mathbf{x})] = \int d^d x \left( \tfrac{1}{2} r \phi^2(\mathbf{x}) + \tfrac{1}{4} u \phi^4(\mathbf{x}) - h\phi(\mathbf{x}) + \tfrac{1}{2} |\boldsymbol{\nabla}\phi|^2 \right), \tag{184}$$

can be expressed in Fourier space using the propagating modes $\psi_{\mathbf{k}}$ in hypercubic systems with PBC's,

$$\phi(\mathbf{x}) = \sum_{\mathbf{k}} \tilde{\phi}_{\mathbf{k}} \psi_{\mathbf{k}} = \frac{1}{\sqrt{V}} \sum_{\mathbf{k}} \tilde{\phi}_{\mathbf{k}} e^{i\mathbf{k}\cdot\mathbf{x}}, \quad \mathbf{k} = \frac{2\pi}{L}\mathbf{n}, \ \mathbf{n} \in \mathbb{Z}^d \tag{185}$$

as

$$F_{\mathrm{GLW}}[\tilde{\phi}_{\mathbf{k}}] = \tfrac{1}{2}\sum_{\mathbf{k}}(r + |\mathbf{k}|^2)|\tilde{\phi}_{\mathbf{k}}|^2 + \tfrac{1}{4}uL^{-d}\sum_{\mathbf{k}_1,\mathbf{k}_2,\mathbf{k}_3}\tilde{\phi}_{\mathbf{k}_1}\tilde{\phi}_{\mathbf{k}_2}\tilde{\phi}_{\mathbf{k}_3}\tilde{\phi}_{-(\mathbf{k}_1+\mathbf{k}_2+\mathbf{k}_3)} - hL^{d/2}\tilde{\phi}_0. \tag{186}$$

It is useful to consider the zero mode $\tilde{\phi}_0$ separately to the non-zero modes $\tilde{\phi}_{\mathbf{k}\neq 0}$. The former contributes to the non-vanishing average order parameter in the ordered phase. The latter do not develop non-zero average values even below the critical temperature (see Section 1). Therefore we write an expansion with essentially two different types of terms: depending or not on the zero mode,

$$\begin{aligned}
F_{\mathrm{GLW}}[\tilde{\phi}_0, \tilde{\phi}_{\mathbf{k}\neq 0}] \ \simeq \ & \tfrac{1}{2}\Big(r + \frac{3u}{2L^d}\sum_{\mathbf{k}\neq 0}|\tilde{\phi}_{\mathbf{k}}|^2\Big)\tilde{\phi}_0^2 + \tfrac{1}{4}\frac{u}{L^d}\tilde{\phi}_0^4 - hL^{d/2}\tilde{\phi}_0 \\
& + \tfrac{1}{2}\sum_{\mathbf{k}\neq 0}(r + |\mathbf{k}|^2)|\tilde{\phi}_{\mathbf{k}}|^2 + \dots
\end{aligned} \tag{187}$$

where we have not explicitly written terms higher than Gaussian for the non zero modes. One can clearly see how the DIV $u$ contaminates the temperature field $r$ to quadratic order in the zero mode expansion while no contamination of the same type appears for the non-zero modes (second line in equation (187)). The behaviour of $\tilde{\phi}_0$ is thus controlled by the GFP modified by the presence of the dangerous irrelevant variable, while the $\tilde{\phi}_{\mathbf{k}\neq 0}$ modes are governed solely by the GFP.

We may therefore infer that the anomalous scaling (which we refer to as Q scaling), like in equations (126) or (127) (this is presented in more detail in Section 9), does not hold for the non-zero modes. Wittmann and Young [46] analyzed several $\mathbf{k} \neq 0$ modes and concluded that the susceptibilities calculated from these modes indeed display *standard* FSS in

$$\chi_{\mathbf{k}\neq 0} = L^d \langle |\tilde{\phi}_{\mathbf{k}\neq 0}|^2\rangle_L \sim L^2 \tag{188}$$

with

$$\tilde{\phi}_{\mathbf{k}} = \frac{1}{\sqrt{V}}\int d^d x\, \phi(\mathbf{x})\, e^{-i\mathbf{k}\cdot\mathbf{x}}. \tag{189}$$

We nevertheless believe that this statement may be misleading, since labeling the power of 2 in equation (188) as *standard* FSS can either refer to Landau scaling, or to the Gaussian Fixed point scaling (see Table 7). The common belief was that *standard* FSS is Landau scaling. Indeed, as far as we know, we were the first to refer explicitly to Gaussian FSS for equation (188), but like others, our understanding of the problem has evolved with time[5] so it is an important matter to be able to distinguish the two situations.

An obvious strategy to discriminate between both scenarios[6] is to analyse the FSS behaviour of the "non-zero modes magnetization" for example. Indeed, we predict for this

---

[5]In our first contribution in this topic [34], we referred to the $\chi_L \sim L^2$ behaviour as *Gaussian* scaling, having in mind MFT or Landau scaling.

[6]Note that both scenarios emerge directly from the action (184) and are therefore implicit in the homogeneous form (99). This single form, along with corrections to be discussed in Sec.5.4 are sufficient to describe scaling and FSS.

quantity a Gaussian FSS

$$m_{\mathbf{k} \neq 0} = \langle |\tilde{\phi}_{\mathbf{k} \neq 0}| \rangle_L \sim L^{-\frac{d-2}{2}} \tag{190}$$

instead of $L^{-1}$ for Landau scaling, and we have checked in Ref. [47] that (190) is indeed correct. We refer to the GFP scaling described in Section 4, which manifests as Eqs (188) and (190), as G scaling in order to distinguish it from the Q scaling associated with zero modes and from standard FSS that comes from the Landau picture in Section 3.

# 8 The Long-range Ising model

The long-range Ising model (LRIM) is a variant of the Ising model in which the value of the upper critical dimension can be controlled by the exponent which governs the decay of the spin-spin interactions. This has been extensively studied in the context of FSS above the upper critical dimension by Luijten and Blöte [38, 39, 48].

The LRIM is defined by the Hamiltonian

$$\mathcal{H} = -\sum_{i,j} J(\mathbf{x}_i - \mathbf{x}_j) s_{\mathbf{x}_i} s_{\mathbf{x}_j} - H \sum_i s_{\mathbf{x}_i} \tag{191}$$

where, as for the ordinary Ising model, the spins, located at positions $\mathbf{x}_i$, take the values $s_{\mathbf{x}_i} = \pm 1$, but the exchange interaction is governed by the algebraic decay

$$J(\mathbf{x}_i - \mathbf{x}_j) = \frac{J}{|\mathbf{x}_i - \mathbf{x}_j|^{d+\sigma}}. \tag{192}$$

The parameter $\sigma$ is free and $\sigma \geq 2$ gives the same universality class as the nearest neighbour model for which $J(\mathbf{x}_i - \mathbf{x}_j) = J\delta(\mathbf{x}_i - \mathbf{x}_j - \boldsymbol{\mu})$ where $\boldsymbol{\mu}$ are the generating vectors of the unit cell. In Fourier space, the counterpart of the expansion (187) is

$$
\begin{aligned}
F_{\mathrm{GLW}}[\tilde{\phi}_0, \tilde{\phi}_{\mathbf{k} \neq 0}] \quad \simeq \quad & \tfrac{1}{2}\Big( r + \frac{3u}{2L^d} \sum_{\mathbf{k} \neq 0} |\tilde{\phi}_{\mathbf{k}}|^2 \Big) \tilde{\phi}_0^2 + \tfrac{1}{4}\frac{u}{L^d} \tilde{\phi}_0^4 - h L^{d/2} \tilde{\phi}_0 \\
& + \tfrac{1}{2} \sum_{\mathbf{k} \neq 0} (r + |\mathbf{k}|^2 + C_\sigma |\mathbf{k}|^\sigma) |\tilde{\phi}_{\mathbf{k}}|^2 + \dots
\end{aligned}
\tag{193}
$$

where the $C_\sigma$ term arises from the Fourier transform of the interaction term in (192).

The RG analysis of the model leads to the GFP anomalous dimensions reported in Table 6. The modified scaling dimensions are obtained following the same lines as in equations (165)

| $x_\phi = \frac{d-\sigma}{2}$ | $y_t = \sigma$ | $y_h = \frac{d+\sigma}{2}$ | $y_u = 2\sigma - d$ |
|---|---|---|---|
| $\alpha_{\mathrm{G}} = 2 - \frac{d}{\sigma}$ | $\beta_{\mathrm{G}} = \frac{d-\sigma}{2\sigma}$ | $\delta_{\mathrm{G}} = \frac{d+\sigma}{d-\sigma}$ | $d_{\mathrm{uc}} = 2\sigma$ |
| $\gamma_{\mathrm{G}} = 1$ | $\nu_{\mathrm{G}} = \frac{1}{\sigma}$ | $\eta_{\mathrm{G}} = 2 + \sigma$ | |

Table 6: Critical exponents at the Gaussian fixed point of the LRIM. The value $\sigma = 2$ gives the ordinary nearest neighbour Ising results ($\phi^4$).

and (166),

$$y_t^* = y_t(\sigma) - \tfrac{1}{2}y_u(\sigma) = \frac{d}{2}, \tag{194}$$

$$y_h^* = y_h(\sigma) - \tfrac{1}{4}y_u(\sigma) = \frac{3d}{4}, \tag{195}$$

and we find the same values as for the case of the nearest-neighbour model. It is clear also that the first line in (193), which controls the zero mode, is the same as in the short-range model.

Our predictions for the FSS behaviour of zero and non-zero modes (we limit ourselves to the magnetization and susceptibility) are thus the following:

$$\text{Q scaling:} \quad m_0 = \langle |\phi_0| \rangle_L \sim L^{-\frac{d}{4}}, \tag{196}$$

$$\chi_0 = L^d \langle |\phi_0|^2 \rangle_L \sim L^{\frac{d}{2}}, \tag{197}$$

$$\text{G scaling:} \quad m_{\mathbf{k} \neq 0} = \langle |\tilde{\phi}_{\mathbf{k} \neq 0}| \rangle_L \sim L^{-\frac{d-\sigma}{2}}, \tag{198}$$

$$\chi_{\mathbf{k} \neq 0} = L^d \langle |\tilde{\phi}_{\mathbf{k} \neq 0}|^2 \rangle_L \sim L^\sigma. \tag{199}$$

These expressions have been carefully checked in Refs. [47] and [40], where LRIM with values of $\sigma = 0.1$ or $0.2$ were studied for 1D and 2D (i.e. in the regime above the upper critical dimension $d_{\text{uc}}(\sigma) = 2\sigma$). Once again the behaviour of the non-zero modes clearly appears to differ from Landau standard scaling, and is governed by the pure GFP exponents instead.

## 9 Dangerous irrelevancy in the correlation sector

Since Fisher's breakthrough forty years ago rescued the RG framework for the free-energy sector in the thermodynamic limit (Sec.5.1) above the upper critical dimension, DIVs were extended to finite-size scaling corrections (Sec.5.4). However, problems still remain and in this section we further build on Fisher's legacy to preserve the RG framework. In particular, we extend DIV's to the correlation sector. In doing so, closely follow Luijten and Blöte's approach that we proposed to generalize in Ref. [47]

### 9.1 The use of ϙ, the rescue of hyperscaling and finite-size scaling

We return firstly to Section 5.3 and BNPY's extension of the DIV mechanism to the free energy itself. As stated, they assumed that the correlation length could obey a similar homogeneity law (140) but insisted that $q_1 = 0$ and left $q_2$ and $q_3$ open. For the free energy in Eq.(138), BNPY gave three reasons why $p_1 = 0$ and hence $d^* = d$. In a series of papers [49] we advocated that the correlation length should be allowed exceed the length — as Brézin had shown for $d = 4$ and general $n$. In particular, we showed that

$$\xi_L \sim L^{\textrm{ϙ}} \tag{200}$$

with

$$\textrm{ϙ} = d/d_{\text{uc}}. \tag{201}$$

Therefore we write

$$\xi_L(t, h, u) = b^{\textrm{ϙ}} \Xi^{\pm}(b^{y_t^+} t u^{q_2}, b^{y_h^+} h u^{q_3}). \tag{202}$$

where

$$\text{ϙ} = 1 + q_1 y_y, \quad y_t^+ = y_t + q_2 y_u, \quad y_h^+ = y_h + q_3 y_u. \tag{203}$$

The value for ϙ in Eq.(201) determines $q_1 = -1/n$. In the thermal sector, matching Eq.(202) with the value $\nu_{\text{MFT}} = 1/2$ in Eq.(85) determines $q_2 = -2/n$, identical to $p_2$. Likewise, matching Eq.(202) with the value $\nu_{\text{c MFT}} = (n-2)/2(n-1)$ in Eq.(85) determines $q_3 = -1/n$, identical to $p_3$. Note that, while the correct correlation-length critical exponent $\nu_{\text{MFT}}$ is identical to that coming from the GFP $\nu_{\text{GFP}} = 1/y_t = 1/2$ in Eq.(47), their counterparts in the magnetic sector do not coincide and $\nu_{\text{c MFT}}$ differs from $\nu_{\text{c GFP}} = 1/y_h$ from Eq.(48). So, technically although $\nu_{\text{MFT}} = 1/y_t$, this is by coincidence only and the correct expression is $\nu_{\text{MFT}} = 1/y_t^+ = 1/y_t^*$. Note also that because $q_2 = p_2$ and $q_3 = p_3$, one has $y_t^+ = y_t^*$ and $y_h^+ = y_h^*$. We henceforth only use the starred scaling dimensions. I.e., to extend Eqs.(144) and (145) to the correlation sector, we write

$$\nu_{\text{MFT}} = \frac{\text{ϙ}}{y_t^*} \quad \nu_{\text{c MFT}} = \frac{\text{ϙ}}{y_h^*}. \tag{204}$$

Equations (147) and (148) can now be written above $d_{\text{uc}}$ in a very consistent manner as

$$d^* = \text{ϙ} d_{\text{uc}}, \quad y_t^*(d) = \text{ϙ} y_t(d_{\text{uc}}), \quad y_h^*(d) = \text{ϙ} y_h(d_{\text{uc}}). \tag{205}$$

Likewise, we believe that there is no reason to invoke a mechanism by which the rescaling of the temperature and the magnetic field in equations (165) and (166) would differ for different physical quantities. A second extension proposed is to use the correct (starred) scaling dimension ($2x_\phi^*$ in fact) of the matter field to describe the dimension of the correlation function. From these considerations, we are led to the following scaling hypotheses (equation (167) is rewritten for the sake of clarity),

$$f_L(t, h, u) = b^{-d} \mathscr{F}^\pm(X, Y, bL^{-1}), \tag{206}$$

$$g_L(\mathbf{x}, t, h, u) = b^{-2x_\phi^*} \mathscr{G}^\pm(b^{-1}\mathbf{x}, X, Y, bL^{-1}), \tag{207}$$

$$\xi_L(t, h, u) = b^{\text{ϙ}} \Xi^\pm(X, Y, bL^{-1}). \tag{208}$$

These are expressed in terms of the two rescaled variables

$$X = b^{y_t^*}(t u^{-2/n} - \tilde{p} u^{(n-2)/n} b^{y_u - y_t}), \tag{209}$$

$$Y = b^{y_h^*} h u^{-1/n}, \tag{210}$$

and with the scaling dimensions

$$y_t^* = d\Big(1 - \frac{2}{n}\Big), \; y_h^* = d\Big(1 - \frac{1}{n}\Big), \; x_\phi^* = \frac{d}{n}, \; \text{ϙ} = \frac{d}{2}\Big(1 - \frac{2}{n}\Big) \tag{211}$$

having used (147) with (91), (92), (93) and extended $x_\phi^*$ from Ref. [47] for the $n = 4$ case, and where $\tilde{p}$ is given in Eq.(158). Obviously these scaling forms apply to the Q-sector only (Fourier $Q$-modes). The G sector is unaffected by DIVs and the finite-size counterparts of Eqs.(34), (36) and (37) apply there. Following a suggestion by Michael Fisher, a new exponent ϙ was introduced in [49] for the Q sector.[7] We like the use of this exponent, since it is very easy

---

[7] We are indebted to Michael Fisher who, having invited us to meet him in the Royal Society just before he retired in 2012, suggested the usage of this nice archaic Greek letter instead of the letter $q$ which we had used hitherto [34]. The notation $q$ itself came from usage of $\hat{q}$ for the exponent governing the logarithmic correction to scaling of the correlation length in [10, 11] and as a nod to BNPY's usage of $q_1$ in Eq.(140) [35].

to translate equations from one universality class to another in terms of $d/d_{uc}$, and also to generalize from the GFP values (which we recover reverting $\varrho$ to 1), so one can also rewrite the exponents (211) in the form

$$y_t^* = 2\varrho, \ y_h^* = \frac{d}{2} + \varrho, \ x_\phi^* = \frac{d}{2} - \varrho, \ \varrho = \frac{d}{d_{uc}}. \tag{212}$$

An interesting use of $\varrho$ is in a new form of the hyperscaling relation. Setting $b = |t|^{-1/y_t}$ in (208) delivers $\xi_\infty \sim |t|^{-\varrho/y_t^*}$, hence $y_t^* = \varrho/\nu$, then in (206) we get $f_\infty \sim |t|^{d/y_t^*} \sim |t|^{2-\alpha}$ which leads to

$$\alpha = 2 - \frac{\nu d}{\varrho}. \tag{213}$$

This repairs[8] the hyperscaling relation above $d_{uc}$, and it holds also below $d_{uc}$ where $\varrho = 1$. (Obviously $\varrho = 1$ for the non-zero or orthogonal Fourier modes in high dimensions too.)

The new exponent already enabled predictions to be more naturally expressed for models such as the nearest-neighbour Ising model, percolation above its critical dimension $d_{uc} = 6$ and for LRIM's with various dimensions above $d_{uc}(\sigma)$ (with periodic boundary conditions) [51,52]. The extension to general values of $n$ is obvious and we collect the predictions for arbitrary $d > d_{uc}$ for quantities which have been discussed above:

$$m_L \sim L^{-(\frac{d}{2}-\varrho)}, \ \chi_L \sim L^{2\varrho}, \ e_L \sim L^{-(d-2\varrho)}, \ c_L \sim L^{4\varrho-d}, \tag{214}$$

$$\xi_L \sim L^\varrho, \ g_L(X_0) \sim L^{-(d-2\varrho)}, \tag{215}$$

$$t_L \sim L^{-2\varrho}, \ \Delta\beta_L \sim L^{-2\varrho}, \ |h^L| \sim L^{-(\frac{d}{2}+\varrho)}, \tag{216}$$

$$h_L^{LY} \sim L^{-(\frac{d}{2}+\varrho)}, \ t_L^F \sim L^{-2\varrho}. \tag{217}$$

Note that all of the above scaling formulas are readily obtained by replacing the standard FSS prescription that converts Eq.(118) to Eq.(120) by the prescription

$$Q_\infty(t,0) \sim |t|^\rho \longrightarrow Q_L(t=0,0) \sim L^{-\varrho\rho/\nu}. \tag{218}$$

This is what we call Q-scaling and it holds at the pseudocritical point as we shall see shortly. With Q-scaling to hand, FSS holds above the upper critical dimension.

The different approaches that we have discussed so far are collected and compared in Table 7 for the $\phi^4$ model with periodic boundary conditions. In this table, the first column lists known results in the thermodynamic limit or for finite-size scaling. The remaining columns correspond to the different approaches that we have described so far, with the symbol $\sqrt{}$ to denote an agreement and, in cases of disagreement, the prediction made by the (incorrect) theory considered is given explicitly. The last column is for Q scaling.

## 9.2 Corrections to scaling and crossover

Equations (214)–(217) are the leading contributions, but corrections to scaling can play important roles as well, and somewhat change the picture (and don't forget the orthogonal G-

---

[8]Since Josephson's inequality $\nu d \geq 2 - \alpha$ was introduced in 1967 [50], literature, including textbooks, on statistical physics, lattice field theory, etc. refer to hyperscaling as "failing" above the upper critical dimension. This statement should no longer be used in statistical physics — hyperscaling does not fail because the RG does not fail above the upper critical dimension. Moreover, the hyperscaling relation should rather be rewritten properly as (213) and not as (30).

| The correct results | Landau scaling[1] | GFP[2] | Fisher DIV[3] | BNPY[4] | ϙ[5] |
|---|---|---|---|---|---|
| $c_\infty(t,0) \sim |t|^0$ | √ | $\alpha_{\mathrm G} = 2 - \frac{d}{2}$  √ | √ | √ | √ |
| $m_\infty(t,0) \sim |t|^{1/2}$ | √ | $\beta_{\mathrm G} = \frac{d-2}{4}$  √ | √ | √ | √ |
| $\chi_\infty(t,0) \sim |t|^{-1}$ | √ | √ | √ | √ | √ |
| $m_\infty(0,h) \sim |h|^{1/3}$ | √ | $\delta_{\mathrm G} = \frac{d+2}{d-2}$  √ | √ | √ | √ |
| $\xi_\infty(t,0) \sim |t|^{-1/2}$ | √ | √ | √ | ? | √ |
| $g_\infty(\mathbf{x},0,0) \sim |\mathbf{x}|^{-(d-2)}$ | √ | √ | √ | ? | √ |
| $\xi_L(t=0,0) \sim L^{d/4}$ [29] | $L$ | $L$ | $L$ | $L$ | √ |
| $\chi_L(t=0,0) \sim L^{d/2}$ [30] | $L^2$ | $L^2$ | $L^2$ | √ | √ |
| $t_L(t=0,0) \sim L^{-d/2}$ [30] | $L^{-1/2}$ | $L^{-1/2}$ | $L^{-1/2}$ | √ | √ |
| $m_L(t=0,0) \sim L^{-d/4}$ | $L^{-1}$ | $L^{1-d/2}$ | $L^{-1}$ | √ | √ |
| $g_L(L/2,t=0,0) \sim L^{-d/2}$ | $L^{-(d-2)}$ | $L^{-(d-2)}$ | $L^{-(d-2)}$ | ? | √ |
| $h_L^{\mathrm{LY}}(t=0) \sim L^{-3d/4}$ | $L^{-3}$ | $L^{-(d+2)/2}$ |  |  | √ |
| $t_L^{\mathrm{F}}(t=0) \sim L^{-d/2}$ | $L^{-2}$ | $L^{-2}$ |  |  | √ |

Table 7: Summary of the evolution of the scaling picture above the upper critical dimension for the $\phi^4$ model. The first column presents the correct results (FSS predictions are for a system with periodic boundary conditions). In the other columns, we give the (incorrect) results predicted when they are different. A question mark means that the quantity hasn't been considered in the corresponding scenario. [1]FSS with Landau exponents, [2]Predictions from the RG eigenvalues at the Gaussian Fixed Point, [3]Corrections made by the scenario of Fisher, [4]Most of the results presented in this column correspond to BNPY's version of the scenario of Fisher and are checked in Ref. [30], [5]Q scaling. of the results presented in the last column are checked in Ref. [34].

modes which are always lurking in the background). The explicit inclusion of corrections to scaling has been introduced analytically and masterfully checked numerically by Luijten [39]. The role of these corrections could a priori be the origin of a discrepancy that we will discover in Section 10 for systems with free boundary conditions (FBC). This option will not turn out to be the right one, as we shall see, but we must logically exploit it and, in any case, it is important to explore the corrections in the vicinity of the dominant behaviour. Rather than assuming different scaling hypotheses for different sets of boundary conditions, universality suggests a single FSS behaviour in the thermodynamic limit that could leave enough room for crossover effects to take place, and this more severely to manifest in systems with FBC's. Anticipating part of the next section, the suspicion in favor of this scenario comes from the numerical results for FBC's which depend very strongly on the manner of carrying out the calculations — keeping a core of spins only, or removing various boundary submanifolds like corners, edges, surfaces, etc. The resulting exponents scatter around values for which it is hard to find a consistent interpretation. Let us quote a recent work by Lundow [53]

> A system with periodic boundary conditions then has $\chi \propto L^{5/2}$. Deleting the $5L^4$ boundary edges we receive a system with free boundary conditions and now $\chi \propto L^2$. In the present work we find that deleting the $L^4$ boundary edges along just one direction is enough to have the scaling $\chi \propto L^2$. It also appears that deleting $L^3$ boundary edges results in an intermediate scaling, here estimated to $\chi \propto L^{2.275}$.

At the risk of repeating ourselves, we will see that this is not the full scenario at work in reality but we cannot rule it out a priori and this is the main reason for this section. We also list tables of the leading and first correction exponents for various quantities and various universality classes and this may be helpful for future studies. So, we will exploit further the formalism introduced previously in Section 5.4.

Let us specify the discussion by presenting the case of the susceptibility. Extracting the finite-size susceptibility from equation (206) in zero magnetic field leads to

$$\chi_L(t,0,u) = L^{2\varrho}\mathscr{X}(X) \tag{219}$$

where the variable $X$ is defined in (209). We call this variable $X_L(t,u)$ for $b = L$, and its explicit form in terms of $d_{\rm uc}$ and $\varrho$ is

$$X_L(t,u) = L^{2\varrho}tu^{-\frac{d_{\rm uc}-2}{d_{\rm uc}}} - \tilde{p}u^{\frac{2}{d_{\rm uc}}}L^{2\varrho+y_u-y_t} \tag{220}$$

with $y_u - y_t = (4 - 2d_{\rm uc}\varrho)/(d_{\rm uc} - 2) < 0$. The first term grows faster with the system size and dominates in the thermodynamic limit for the values of $d > d_{\rm uc}$ considered here.

At the pseudo-critical point, as defined in (169), the scaling variable takes the value $X_L(t_L) = X_0$ and the susceptibility is simply

$$\chi_L(t_L,0,u) \simeq \mathscr{X}(X_0)L^{2\varrho}, \tag{221}$$

with the correct FSS exponent.

Let us suppress the $u$ dependence and consider a simplified expression $X_L(t)$ for our forthcoming discussion:

$$X_L(t) = AL^{2\varrho}t - BL^{4(1-\varrho)/(d_{\rm uc}-2)}. \tag{222}$$

At the critical point $t = 0$, $X_L(t=0) = -BL^{y_{\rm corr}}$ and

$$\chi_L(0,0,u) \simeq \mathscr{X}(-BL^{y_{\rm corr}})L^{2\varrho} \tag{223}$$

with $y_{\rm corr} = y_u - y_t + y_t^* = 4(1-\varrho)/(d_{\rm uc} - 2) < 0$. The scaling function is regular with an argument $X_L(t=0) \to 0$ when $L \to \infty$ which allows to expand $\mathscr{X}(-BL^{y_{\rm corr}})$ to first order in the vicinity of $X_L(t=0) = 0$,

$$\chi_L(0,0,u) \simeq \mathscr{X}(t=0)L^{2\varrho} - B\mathscr{X}'(t=0)L^{2\varrho+y_{\rm corr}}+\dots, \tag{224}$$

where now $2\varrho + y_{\rm corr} = (2\varrho(d_{\rm uc} - 4) + 4)/(d_{\rm uc} - 2)$. It is very clear that here a crossover takes place, depending on the relative magnitudes of both terms. If the scaling function is such that for small enough sizes the condition

$$(L/\ell_0)^{4(1-\varrho)/(d_{\rm uc}-2)} \equiv |B\mathscr{X}'(t=0)/\mathscr{X}(t=0)|L^{4(1-\varrho)/(d_{\rm uc}-2)} \gg 1 \tag{225}$$

holds (remember that $4(1-\varrho)/(d_{\rm uc} - 2) < 0$), the leading behaviour is governed in this regime by the second term. Instead of $2\varrho$, the FSS exponent measured there becomes closer to the corrected value $(2\varrho(d_{\rm uc} - 4) + 4)/(d_{\rm uc} - 2)$. In the case where $d_{\rm uc} = 4$, such as in the Ising model, this leads to $\chi_L(t=0) \sim L^2$ — precisely the same as what would arise from Landau FSS. This cannot hold in the thermodynamic limit, so that this is only an effective exponent, and $2\varrho$ remains the only true FSS exponent for the susceptibility there.

The leading corrections to scaling have been studied in the thesis of Flores-Sola [40], but equation (224) is only the beginning of the expansion, including all sorts of corrections to scaling [54], derived and checked by Luijten [39], e.g.

$$
\begin{aligned}
\chi_L(t,u) = L^{2y_h^*-d}(a_0 \quad &+ \quad a_1 L^{y_t^*}[t(1+s_1 L^{y_u}) + p_1 L^{y_u - y_t}] \\
&+ \quad a_2 L^{2y_t^*}[t(1+s_1 L^{y_u}) + p_1 L^{y_u-y_t}]^2 + \ldots \\
&+ \quad b_1 L^{y_u} \\
&+ \quad b_2 L^{2y_u} + \ldots \quad ).
\end{aligned}
\tag{226}
$$

| Model | $\phi^n$ | $m_L(0,0,u)$ | | $\chi_L(0,0,u)$ | | $c_L(0,0,u)$ | |
|---|---|---|---|---|---|---|---|
| | | $y_h^* - d$ | $y_h^* - d + y_{\mathrm{corr}}$ | $2y_h^* - d$ | $2y_h^* - d + y_{\mathrm{corr}}$ | $2y_t^* - d$ | $2y_t^* - d + y_{\mathrm{corr}}$ |
| Magnets, SAW | $\phi^4$ | $\varphi - \frac{d}{2}$ | $2 - \frac{d}{2} - \varphi$ | $2\varphi$ | $2$ | $4\varphi - d$ | $2\varphi - d + 2$ |
| Percolation | $\phi^3$ | $\varphi - \frac{d}{2}$ | $1 - \frac{d}{2}$ | $2\varphi$ | $1 + \varphi$ | $4\varphi - d$ | $3\varphi - d + 1$ |
| Tricriticality | $\phi^6$ | $\varphi - \frac{d}{2}$ | $4 - \frac{d}{2} - 3\varphi$ | $2\varphi$ | $4 - 2\varphi$ | $4\varphi - d$ | $4 - d$ |

Table 8: Summary of the crossover expected for FSS at the asymptotic critical point for the magnetization, the susceptibility and the specific heat for the $\phi^n$ models. In this table, $y_{\mathrm{corr}} = y_u - y_t + y_t^* = 4(1 - \varphi)/(d_{\mathrm{uc}} - 2)$.

Similar crossovers are obtained for the other thermodynamic quantities. In Table 8 we collect the exponents of the leading and first correction term for the magnetization, the susceptibility and the specific heat for the different universality classes that we usually consider in this paper.

## 9.3 Corrections in the correlation sector

The case of the correlations can be discussed in a similar manner. The assumption (207), rewritten here in complete form in zero magnetic field reads as

$$
g_L(\mathbf{x}, t, 0, u) = b^{-2x_\phi^*}\mathscr{G}^\pm(b^{-1}\mathbf{x}, b^{y_t^*}tu^{-\frac{2}{n}} - \tilde{p}u^{1-\frac{2}{n}}b^{y_u - y_t + y_t^*}, bL^{-1}).
\tag{227}
$$

Using the same arguments as above, one can fix e.g. $b = |\mathbf{x}|$ to get a first-order expansion at the pseudo-critical point $t_L$ (unit vector $\mathbf{u} = \mathbf{x}/|\mathbf{x}|$ omitted)

$$
\begin{aligned}
g_L(\mathbf{x}, t_L, 0, u) &= |\mathbf{x}|^{-2x_\phi^*}\mathscr{G}^\pm(X_0, |\mathbf{x}|L^{-1}) \\
&\simeq |\mathbf{x}|^{-2x_\phi^*}[\mathscr{G}^\pm(X_0, 0) + |\mathbf{x}|L^{-1}\mathscr{G}^{\pm(0,0,1)}(X_0, 0)],
\end{aligned}
\tag{228}
$$

and at the asymptotic critical point,

$$
\begin{aligned}
g_L(\mathbf{x}, 0, 0, u) &= |\mathbf{x}|^{-2x_\phi^*}\mathscr{G}^\pm(-B|\mathbf{x}|^{y_{\mathrm{corr}}}, |\mathbf{x}|L^{-1}) \\
&\simeq |\mathbf{x}|^{-2x_\phi^*}[\mathscr{G}^\pm(0,0) - B|\mathbf{x}|^{y_{\mathrm{corr}}}\mathscr{G}^{\pm(0,1,0)}(0,0) + |\mathbf{x}|L^{-1}\mathscr{G}^{\pm(0,0,1)}(0,0)].
\end{aligned}
\tag{229}
$$

Here, $\mathscr{G}^{(0,1,0)}$ for example, means that we take the first derivative of the function $\mathscr{G}$ wrt its second argument.

It is probably easier to study numerically a fixed ratio of the distance $|\mathbf{x}|$ to the size of the system, e.g. $\frac{1}{2}$. Then one chooses $b = L$ and it comes out

$$
\begin{aligned}
g_L(L/2, t_L, 0, u) &= L^{-2x_\phi^*}\mathscr{G}^\pm(\tfrac{1}{2}, X_0, 0), &(230)\\
g_L(L/2, 0, 0, u) &= L^{-2x_\phi^*}\mathscr{G}^\pm(\tfrac{1}{2}, -BL^{y_{\text{corr}}}, 0) \\
&\simeq L^{-2x_\phi^*}[\mathscr{G}^\pm(\tfrac{1}{2}, 0, 0) - BL^{y_{\text{corr}}}\mathscr{G}^{\pm(0,1,0)}(\tfrac{1}{2}, 0, 0)]. &(231)
\end{aligned}
$$

While (230) exhibits a pure decay with an exponent $-2x_\phi^*$, (231) should display a crossover between an exponent $-2x_\phi^* + y_{\text{corr}}$ at small sizes to $-2x_\phi^*$ in the thermodynamic limit. We report the corresponding values in Table 9 for the three usual universality classes.

| Model | $\phi^n$ | $2x_\phi^*$ | $\eta^*$ | $2x_\phi^* - y_{\text{corr}}$ | $\eta_{\text{corr}}$ |
|---|---|---|---|---|---|
| | | | | $g_L(L/2, 0, 0, u)$ | |
| Magnets, SAW | $\phi^4$ | $d - 2\varphi$ | $2 - 2\varphi$ | $d - 2$ | $0$ |
| Percolation | $\phi^3$ | $d - 2\varphi$ | $2 - 2\varphi$ | $d - 1 - \varphi$ | $1 - \varphi$ |
| Tricriticality | $\phi^6$ | $d - 2\varphi$ | $2 - 2\varphi$ | $d - 4 + 2\varphi$ | $2\varphi - 2$ |

Table 9: Summary of the crossover expected at the asymptotic critical point for the correlation function for the $\phi^n$ models. The corresponding values of $\eta$ exponents, defined by $2x_\phi^* = d - 2 + \eta^*$ and $2x_\phi^* - y_{\text{corr}} = d - 2 + \eta_{\text{corr}}$ are also given.

As far as we know, the correlation function crossover in terms of distance has not been numerically checked yet.

# 10  The case of Free Boundary Conditions

## 10.1  The problem

We will now devote a moment to the difficult case of free boundary conditions.

To discuss first the easy part, the study of FSS properties of all physical quantities evaluated *at the pseudo-critical point* for FBC's agrees with all what was said previously for systems with PBC's. We believe that there is now a consensus on this, but what is happening right at the critical point is subject to a lot of discussion and consensus has not yet been obtained. In particular, the energy sector is actually misunderstood as we will see.

In Ref. [34], 5D systems of moderate sizes (up to $L \simeq 30$) were studied at $T_c$. Besides the full lattice, to mitigate the role of boundary submanifolds which effectively have lower dimensionalities, outer half of the sites were removed in each direction, leaving a core which is genuinely five-dimensional. The results for the FSS of the susceptibility and the magnetization were not completely conclusive,

$$
\begin{aligned}
\text{core sites} \quad \chi_{\text{core}}(T_c) &\sim L^{1.92}, &(232)\\
m_{\text{core}}(T_c) &\sim L^{-1.575}, &(233)\\
\text{all sites} \quad \chi_{\text{all}}(T_c) &\sim L^{1.71}, &(234)\\
m_{\text{all}}(T_c) &\sim L^{-1.70}. &(235)
\end{aligned}
$$

As we said in Ref. [34],

> *again, at $T_c$, the data follow neither the Gaussian nor the Q-behaviour*

(Q-behaviour referring there to Q scaling). We had then proposed that the asymptotic regime was not reached and that the 5D behaviour was contaminated by the 4D one of the free surfaces and even by edges, corners, etc. of still lower dimensionalities. The perspective offered by the previous section is a tempting solution of the problem. Indeed, the exponent in Eq. (235), for example, is close to the value $2 - \frac{d}{2} - \varphi = -1.75$ for the correction exponent of the magnetization in Table 8. On the other hand, the corresponding exponent in Eq. (234) does not appear to be compatible with the value 2 reported for the correction in the same table. For this very quantity, the core sites susceptibility (232) displays a closer exponent, and that of the magnetization (233) is now right between $\varphi - \frac{d}{2} = -1.25$ and $2 - \frac{d}{2} - \varphi = -1.75$, the leading and correction exponents (see Table 9), however, it seems difficult to reach any reliable conclusion on the basis solely of these results.

Lundow and Markström are experts in simulations and they have studied this problem intensively, reaching far larger systems (up to $L = 160$ in [55]) in which the fraction of "non-bulk" sites is much smaller. They obtained very accurately leading behaviours and corrections to scaling:

$$\chi_L(T_c) = 0.817\, L^2 + 0.083\, L, \tag{236}$$

$$m_L(T_c) = 0.230\, L^{-3/2} + 1.101\, L^{-5/2} - 1.63\, L^{-7/2}. \tag{237}$$

On the basis of these results, Lundow and Markström concluded in favor of a *standard FSS behaviour* of the susceptibility at the critical temperature $\chi_L(T_c) \sim L^2$. The same conclusion, albeit with much lower accuracy, was reached in [34] on the basis of (232). There, the use of the word *Gaussian* was misleading, since we referred in fact to FSS with Landau exponents, $\chi_L \sim L^{\gamma_{\mathrm{MFT}}/\nu_{\mathrm{MFT}}}$, a conclusion that we will see is *not* correct, although $L^2$ is (accidentally) correct.

The case of the magnetization in (237) is a bit more subtle, because the leading exponent $-\frac{3}{2}$ *is not* the standard Landau FSS (this would be $-\beta_{\mathrm{MFT}}/\nu_{\mathrm{MFT}} = -1$). Even more puzzling are the cases of the internal energy and specific heat at criticality, for which the corrections to scaling were also reported in [55]:

$$e_L(T_c) = 0.68 - 1.01L^{-1} + 0.39L^{-3/2}, \tag{238}$$

$$c_L(T_c) = 14.69 - 14.93L^{-1/3}, \tag{239}$$

but we are still lacking an explanation for these results.

In order to clarify the situation, Wittmann and Young [46, 56], and then Flores-Sola et al. [47] considered the behaviour of the Fourier modes in a finite system with free boundary conditions. Following Ref. [57][9], a sine-expansion of the scalar field in the $\phi^4$ action is performed with the boundary conditions $\phi(\mathbf{x}) = 0$ at the free surfaces.

$$\phi(\mathbf{x}) = \sum_{\mathbf{k}} \tilde{\phi}_{\mathbf{k}} \psi_{\mathbf{k}} = \sum_{\mathbf{k}} \tilde{\phi}_{\mathbf{k}} \prod_{\alpha=1}^{d} \sqrt{2/L} \sin k_\alpha x_\alpha, \tag{240}$$

where the wave vector components take the values $k_\alpha = n_\alpha \pi/(L+1)$, $n_\alpha = 1, 2, \ldots, L$.

---

[9]An "underappreciated paper" as described by Wittmann and Young.

The action takes in $\mathbf{k}$−space a form different to that for PBC's. Distinction should be made between the modes for which all $n_\alpha$-values are odd integers, which are analogous to the zero mode in the PBC case (we denote in FBC's their set by $\mathcal{Q}$), and all other modes. We discriminate the modes in $\mathcal{Q}$ because these are the only ones to have the symmetry of the average order parameter wrt the center of the lattice for FBC systems. Therefore, these modes may possibly have a substantial contribution to the equilibrium magnetization. The remaining modes, for which symmetry reasons exclude any substantial contribution to the average order parameter, are denoted by $\mathcal{G}$ (like *Gaussian*). The action now reads as [57]

$$F_{\text{GLW}}[\tilde{\phi}_{\mathbf{k}}] = \frac{1}{2} \sum_{\mathbf{k}} \left( r_0 + c|\mathbf{k}|^2 \right) \tilde{\phi}_{\mathbf{k}}^2 - \left(\frac{8}{L}\right)^{\frac{d}{2}} h \sum_{\mathbf{k} \in \mathcal{Q}} \tilde{\phi}_{\mathbf{k}} \prod_{\alpha=1}^{d} \frac{1}{k_\alpha}$$
$$+ \frac{u}{L^d} \sum_{\mathbf{k}_1, \mathbf{k}_2, \mathbf{k}_3, \mathbf{k}_4} \Delta_{\mathbf{k}_1, \mathbf{k}_2, \mathbf{k}_3, \mathbf{k}_4} \tilde{\phi}_{\mathbf{k}_1} \tilde{\phi}_{\mathbf{k}_2} \tilde{\phi}_{\mathbf{k}_3} \tilde{\phi}_{\mathbf{k}_4}. \tag{241}$$

The quantities $\Delta_i$'s are momentum-conserving factors. Two important differences between Eqs. (187) and (241) are observed. The first one is in the quadratic terms. This is the source for the different scaling observed between the two types of boundary conditions. It also appears that the quartic term in Eq. (241) is dangerous for the modes $\mathbf{k} \in \mathcal{Q}$ only (due to the restricted summation $\sum_{\mathbf{k} \in \mathcal{Q}}$) which couple to $h$. We henceforth refer to modes for which $u$ is dangerous (in particular, the zero mode for PBC's and modes with all $n_\alpha$ odds for FBC's) as Q-modes and the remaining ones as G-modes or Gaussian modes.

With the notation $\tilde{m}_{\mathbf{k}} = \tilde{\phi}_{\mathbf{k}}$, already introduced to represent the contribution of a single mode $\mathbf{k}$ to the average magnetization, one defines the corresponding susceptibility by[10]

$$\chi_{\mathbf{k}} = L^d(\langle \tilde{m}_{\mathbf{k}}^2 \rangle - \langle |\tilde{m}_{\mathbf{k}}| \rangle^2). \tag{242}$$

The equilibrium magnetization $m$ takes all modes into account and the total susceptibility is defined accordingly. Wittmann and Young confirmed the scaling at $T_c$ for the total susceptibility (for FBC). They found that the single mode susceptibility also obeys a *standard FSS behaviour*, $\chi_{\mathbf{k}} \sim L^2$ for the modes which *will not acquire a nonzero magnetization*, namely with the smallest wave-vector with an even $n_\alpha$. We will come back to this in a moment.

An argument in favor of this result, given in [46], follows from (almost) ordinary scaling and helps to understand the sense of the word *standard* FSS for the authors. Actually, Wittmann and Young proposed to use $\bar{t} = T - T_L$, with $\bar{t} = t + \text{const} \times L^{-\lambda}$, as the temperature-like scaling variable and the starred RG dimensions in the scaling hypothesis for the susceptibility. They wrote then $\chi_L(\bar{t}) = L^{2y_h^* - d} \mathscr{X}(L^{y_t^*} \bar{t})$ in zero magnetic field and the compatibility with the bulk behaviour $\chi_\infty(t) \sim |t|^{-\gamma_{\text{MFT}}}$ is recovered in the thermodynamic limit by the demand that the asymptotic regime obeys $\mathscr{X}(x) \sim x^{-\gamma_{\text{MFT}}}$ (with $\gamma_{\text{MFT}} = 1$ here) for $x \to \infty$, giving $\chi_L(\bar{t}) \sim L^{2y_h^* - d + y_t^*}(\bar{t})^{-1}$. When $\lambda = d/2$, as it is the case in PBC, this leads at criticality to $\chi_L(T_c) = L^{d/2}$. On the other hand if $\lambda = 2$ (FBC, a point that we comment further below), this amounts to $\chi_L(T_c) = L^2$.

## 10.2 Four possible scenarios

We can however ask why $y_t^*$ and $y_h^*$ have been used in FBC instead of the unstarred exponents, or rather, which ones have been used in fact. Indeed, as we have already noted several times,

---

[10]Actually, for the modes which do not contribute to the average magnetization, (i.e. such that $\langle |\tilde{m}_{\mathbf{k}}| \rangle = 0$), the susceptibilities are defined solely by the first terms in equation (242).

one cannot disentangle, with the susceptibility, the predictions of *standard* (or Landau) FSS from those of the *Gaussian Fixed point* FSS. The susceptibility is clearly not a good quantity to analyse, and the primary aim of Ref. [47] was to test the more discriminating case of the magnetization. The argument of Wittmann and Young for the magnetization would lead to $m_L(T_c) = L^{-\lambda/2}$, hence $m_L(T_c) = L^{-d/4}$ in PBC and $m_L(T_c) = L^{-1}$ in FBC. The argument is thus equivalent to *Landau scaling*, because for FBC, $\lambda = 1/\nu_{\mathrm{MFT}}$.

So we are led to the point where one essentially faces four distinct hypotheses among which one has to discriminate (see Table 10).

| Hypothesis | Scaling of $\chi_L(T_c)$ | Scaling of $m_L(T_c)$ |
|---|---|---|
| 1. Landau (or *standard*) scaling | $L^{\frac{\gamma_{\mathrm{MFT}}}{\nu_{\mathrm{MFT}}}}$ | $L^{-\frac{\beta_{\mathrm{MFT}}}{\nu_{\mathrm{MFT}}}}$ |
| 2. Q scaling | $L^{2\varrho}$ | $L^{\varrho-d/2}$ |
| 3. G scaling | $L^2$ | $L^{1-d/2}$ |
| 4. Crossover | $A_\chi L^{2\varrho} - B_\chi L^{2\varrho+\frac{4(1-\varrho)}{d_{\mathrm{uc}}-2}}$ | $A_m L^{\varrho-d/2} - B_m L^{\varrho-d/2+\frac{4(1-\varrho)}{d_{\mathrm{uc}}-2}}$ |

Table 10: The four scenarios for the FSS of the susceptibility and the magnetization at $T_c$. There, $\varrho = d/d_{\mathrm{uc}}$, $d_{\mathrm{uc}}$ and the $_{\mathrm{MFT}}$ exponents take their respective values for the three different universality classes under consideration.

In the case of the $\phi^4$ universality class, the options are thus

— prediction 1 (Landau scaling) leads to $\chi_L(T_c) \sim L^2$ and $m_L(T_c) \sim L^{-1}$,

— prediction 2 (Q scaling) to $\chi_L(T_c) \sim L^{d/2}$ and $m_L(T_c) \sim L^{-d/4}$,

— prediction 3 (G scaling) to $\chi_L(T_c) \sim L^2$ and $m_L(T_c) \sim L^{-(d-2)/2}$, and

— prediction 4 (crossover to Q scaling) with effective exponents between $d/2$ and 2 for $\chi_L$ and between $-d/4$ and $2 - 3d/4$ for $m$.

Option 2 is clearly ruled out at $T_c$ by the results of Lundow and Markström in (236) (as well as by the results of Wittmann and Young). Options 1, 2 and 4 are all compatible with the results measured for the susceptibility. For the 5D model, it is easy to discriminate, with the magnetization, between option 1 (which predicts the value $-1$), option 3 (prediction $-1.5$) and option 4 (prediction between $-1.25$ and $-1.75$). Equation (237) is clearly in favor of option 3, but does not rule out option 4.

## 10.3 Towards a GFP scaling at $T_c$

In Ref. [47] the magnetization of the LRIM was investigated to tune the parameter $\sigma$ of the interaction decay. In the thesis of Flores-Sola [40] results are reported for various values of $\sigma$ and of $d$. Some of these are collected in Table 11 for the magnetization at $T_c$ and at $T_L$. The expectation from GFP scaling is an exponent $-\beta_{\mathrm{G}}/\nu_{\mathrm{G}}$ with $\beta_{\mathrm{G}} = (d-\sigma)/(2\sigma)$ and $\nu_{\mathrm{G}} = 1/\sigma$, while Q scaling predicts an exponent $-\varrho\beta_{\mathrm{MFT}}/\nu_{\mathrm{MFT}} = -\varrho$.

Other quantities (temperature shift, correlation length, correlation function) are also reported in [40] (and partially collected here) and, although perfectible, all numerical results at $T_c$ support option 3 above (and confirm Q scaling at $T_L$).

| | | $m_L(T_c)$ | $\frac{d-\sigma}{2}$ | $m_L(T_L)$ | $\frac{\wp\beta_{\mathrm{MFT}}}{\nu_{\mathrm{MFT}}}=\frac{d}{4}$ |
|---|---|---|---|---|---|
| $d=1$ | $\sigma=0.1$ | $L^{-0.450(4)}$ | 0.45 | $L^{-0.233(4)}$ | 0.25 |
| | $\sigma=0.2$ | $L^{-0.401(3)}$ | 0.40 | $L^{-0.230(4)}$ | 0.25 |
| $d=2$ | $\sigma=0.1$ | $L^{-0.949(1)}$ | 0.95 | $L^{-0.501(2)}$ | 0.50 |
| | $\sigma=0.2$ | $L^{-0.897(1)}$ | 0.90 | $L^{-0.494(1)}$ | 0.50 |
| | | $\chi_L(T_c)$ | $\sigma$ | $\chi_L(T_L)$ | $\frac{\wp\gamma_{\mathrm{MFT}}}{\nu_{\mathrm{MFT}}}=\frac{d}{2}$ |
| $d=1$ | $\sigma=0.1$ | $L^{0.099(1)}$ | 0.1 | $L^{0.522(3)}$ | 0.5 |
| | $\sigma=0.2$ | $L^{0.200(1)}$ | 0.2 | $L^{0.525(5)}$ | 0.5 |
| $d=2$ | $\sigma=0.1$ | $L^{0.094(2)}$ | 0.1 | $L^{0.985(2)}$ | 1. |
| | $\sigma=0.2$ | $L^{0.198(2)}$ | 0.2 | $L^{0.994(2)}$ | 1. |
| | | $\xi_L(T_c)$ | 1 | $\xi_L(T_L)$ | $\wp=\frac{d}{2\sigma}$ |
| $d=1$ | $\sigma=0.1$ | $L^{1.01(3)}$ | 1 | $L^{4.03(7)}$ | 5 |
| | $\sigma=0.2$ | $L^{1.03(2)}$ | 1 | $L^{2.21(4)}$ | 2.5 |
| $d=2$ | $\sigma=0.1$ | $L^{1.07(7)}$ | 1 | $L^{7.48(4)}$ | 10 |
| | $\sigma=0.2$ | $L^{0.95(6)}$ | 1 | $L^{3.97(4)}$ | 5 |
| | | $g_L(T_c,\lvert\mathbf{x}\rvert=\frac{L}{2})$ | $d-2+\eta_{\mathrm{G}}=d-\sigma$ | $g_L(T_L,\lvert\mathbf{x}\rvert=\frac{L}{2})$ | $d-2+\eta_{\wp}=\frac{d}{2}$ |
| $d=1$ | $\sigma=0.1$ | $L^{-0.86(6)}$ | 0.9 | $L^{-0.487(5)}$ | 0.5 |
| | $\sigma=0.2$ | $L^{-0.83(6)}$ | 0.8 | $L^{-0.483(6)}$ | 0.5 |
| $d=2$ | $\sigma=0.1$ | $L^{-2.07(9)}$ | 1.9 | $L^{-0.954(3)}$ | 1 |
| | $\sigma=0.2$ | $L^{-1.70(9)}$ | 1.8 | $L^{-0.974(3)}$ | 1 |

Table 11: FSS of the magnetization (upper pannel) for the LRIM in $d=1$, 2 and 3 with FBC's at $T_c$ and at $T_L$. The results at the critical temperature support G scaling ($\frac{d-\sigma}{2}$) while those at the pseudo-critical temperature support Q scaling ($\wp\beta_{\mathrm{MFT}}/\nu_{\mathrm{MFT}}$). The three other tables collect results for the susceptibility, the correlation length and the correlation function.

 

The picture now is the following: *Either* we study physical quantities which are related to the Q modes or those related to the G modes. In the first case, which is expected at $T_L$, the DIV has to be taken into account and Q scaling rules (option 2 in Table 10). In the second case, which holds at $T_c$, $u$ is not dangerous and the physics is controlled by the GFP (option 3 in Table 10). The exponents there are those collected in Table 3, *and not Landau exponents* collected in Table 2. There still remains to understand why quantities at $T_c$ averaged over all modes in FBC obey G scaling rather than Q scaling. This is surprising, since average properties are dominated by Q modes. We believe that the reason is entirely due to the behaviour of the shift of the pseudo-critical temperature $t_L = T_c - T_L$. It was indeed shown already in Ref. [57] that

$$t_L \sim L^{-2} \tag{243}$$

for the Ising model with FBC's above its upper critical dimension, while $t_L \sim L^{-d/2}$ for

PBC's. In both cases, the rounding is still governed by the exponent $d/2$, hence we observe that in FBC's, the shift is much larger than the rounding, and $T_c > T_L$. This means that at $T_c$, the finite system is effectively disordered. As a consequence, the average order parameter profile *at $T_c$* is essentially vanishing and Q modes, like G modes have negligible contributions at that temperature, like in the high temperature phase. In a sense, the variable $u$ is not able to render the zero mode order parameter dominant.

This could be the end of the story but for the fact that the above interpretations in the energy sector (internal energy and specific heat) don't match the recent numerical observations of Lundow and Markström in (236). These measurements disagree with all three options: Q scaling, G scaling and also the less likely Landau scaling. The fourth option of a crossover does not do the job either. For the internal energy, it would predict a competition between terms in $L^{-2.5}$ and $L^{-3}$, and for the specific heat, $L^0$ would compete with $L^{-0.5}$. Of course the numerical determination of the internal energy and of the specific heat is known to be made difficult by the presence of important regular contributions, so there is still work to do there.

A last point to emphasize concerns the case of percolation for which also competing theories have been elaborated. Let us remind the reader that the analog of the susceptibility for percolation is the average size of finite clusters $S(p)$ (with $p$ the probability that a site or a bond is occupied) and that a central quantity which plays the role of the free energy density is the density of finite clusters $K(s,p)$. The phase transition which stands out the two phases (existence vs non existence of spanning cluster(s)) occurs at a probability $p_\infty$ and the analog of the reduced temperature there is $\epsilon = |p - p_\infty|$. The homogeneous form of the free energy density then takes the standard form $K(s,p) = b^{-d}\mathscr{F}(\kappa b^{D_f}, \epsilon b^{1/\nu})$. Here, the analog of the magnetic field is $\kappa = s^{-1} - (s^{\max})^{-1}$ with $s^{\max}$ the typical mass of the largest clusters which scales near $p_\infty$ like $s_\infty^{\max}(p) \sim \epsilon^{-\frac{1}{\sigma}} \sim (\xi_\infty)^{D_f}$ with a fractal dimension $D_f$, hence $\sigma = 1/(\nu D_f)$. $D_f$ is the analog of $y_h$ in our discussion.

Above $d_{\mathrm{uc}} = 6$, Antonio Coniglio proposed a scenario according to which there is proliferation of interpenetrating spanning clusters [58], but $D_f$ is stuck to its value $D_{\mathrm{uc}} = 4$ at the upper critical dimension. The resulting free energy density for finite systems takes the form

$$K_L(s,p) = b^{-(d-X)}\mathscr{F}(b^{D_{\mathrm{uc}}}\kappa, b^{1/\nu_c}\epsilon, bL^{-1}) \tag{244}$$

with $X = d - d_{\mathrm{uc}}$ the exponent which measures the proliferation of spanning clusters. According to the literature (see e.g. [59]), this situation is encountered in FBC's. It corresponds exactly to scenario 1 in Table 10, i.e. Landau FSS, a case which was excluded for the IM universality class! Finite systems with PBC's on the other hand seem to obey a different ansatz, with no proliferation of percolating (wrapping) clusters of fractal dimension $D_f = D^* = 2d/3$ [60]. This agrees with scenario 2 of Q FSS in Table 10.

Clearly, progress has been made, but the case of FBC's is not yet fully understood and further studies are underway [61].

# 11 Conclusions

Because they are now endemic in studies of phase transitions and critical phenomena, it is almost forgotten that the notation for the six main critical exponents at one point required standardisation. It was Michael Fisher who assigned the labels $\alpha$, $\beta$, $\gamma$, $\delta$, $\eta$ and $\nu$ to the

observables listed in Eqs.(25–29) in the 1960's [62]. These are linked by the four scaling relations (30)-(33), the last of which was often considered to fail above the upper critical dimension. The alluring simplicity of the (correct) mean-field description of scaling there hid subtleties that appeared to undermine the renormalization group itself, accurately described in Ref. [63], for example, as "plagued by a number of persistent problems".

In the 1980s, Michael Fisher took the first and most important steps to rescue the situation when he identified "the only flaw in the original argument was a failure to recognize and allow for possible singular behaviour of the scaling function." In the spirit of Occam's razor, and fixing only that which appeared to be broken, he addressed the free energy sector in the thermodynamic limit, identifying some irrelevant variables as dangerous. While "the renormalization group framework ha[d] been preserved intact", hyperscaling was sacrificed, its lack of ubiquity only a relatively mild discomfort.

Finite-size scaling had more enigmatic undertones in high dimensions, however. To quote some of the leading protagonists at the turn of the century [64] "although ...all exponents are known, ...and in principle very complete analytical calculations are possible, the existing theories clearly are not so good." A number of ingenious and even elegant formalisms had been developed, reflecting "the progress and setbacks inherent to the evolution of science" alluded to above. Notable amongst these were BNPY's extension of Fisher's dangerous concepts to finite systems [35]; Binder's thermodynamic length [30]; Coniglio's proliferating spanning clusters in percolation [58] and Luijten's and Blöte's [39] inclusions of corrections to scaling. The extension of these concepts to the correlation sector led to the introduction of a new exponent ϙ [49] which, like the story outlined above, has an enigmatic character; although it is a finite-size concept it is needed to recover hyperscaling for the thermodynamic limit as Eq.(213).

However, the puzzles of FSS above the upper critical dimension are still not fully resolved and there is plenty of room for further explorations. We have essentially two different types of behaviour which are partially controlled by the value of the order parameter. The two behaviours are governed either by the pure Gaussian Fixed Point (G scaling), or by the same fixed point, but contaminated by the Dangerous Irrelevant Variable (Q scaling). For physical quantities which do not depend on this variable, all exponents are ordinary Gaussian exponents, and FSS is like $Q_L \sim L^{-\rho_{\mathrm{G}}/\nu_{\mathrm{G}}}$. For other quantities, affected by the DIV, scaling is either like $Q_L \sim L^{-\rho_{\mathrm{G}}/\nu_{\mathrm{G}}}$, or like $Q_L \sim L^{-\varphi \rho_{\mathrm{MFT}}/\nu_{\mathrm{MFT}}}$. The second case is more generic, but when the order parameter (a priori "contaminated") is strongly bounded to zero by the boundary conditions, there is no room for the DIV to develop and the first case is observed. This is what happens specifically at $T_c$ in systems with FBC's. However, this discussion should be moderated by the case of percolation which does not yet fully fit in this picture [61].

As stated, this is not a review. Nor is it a prediction of the future. It is the story of what we consider as being the most important developments in the adaption of RG for high dimensions. Still, having exposed the seemingly inert mean-field realm of high dimensions as a rich ground for fundamental research, we can anticipate new, important and fascinating developments in the future. E.g., while we were finishing these lecture notes, one such paper appeared concerning the extension of the Q FSS formalism to quantum phase transitions [65]. Suffice to say that hyperscaling can no longer be described as "failing" above the upper critical dimension. Instead it is rescued by the emergence of ϙ which can take its place amongst the pantheon of critical exponents that were christened by the same authoritative figure to which this paper is dedicated.

## Acknowledgements

We would like to thank Emilio Flores-Sola, former student of B.B. and R.K., who made his PhD in co-advisory between Coventry University and the Université de Lorraine.

As we finished preparing this manuscript, we learned of the passing of Michael Fisher through the greatest phase transition of them all. We take the opportunity to express our thanks for all he has done to enlighten all of us from the very start to the very end of his long and fruitful career.

We take this opportunity to express our indignation at the Russian invasion of Ukraine and the distorted view of history that preceded it. We refer the reader to Ref. [66] for an analysis of medieval Ukrainian literature which refutes that view and concludes: "Thus the Kyiv cycle of east Slavic epic narratives falls nicely within the European tradition in network terms — it is in many ways like Ireland's heroic tradition and Iceland's social ones." Finally, express our solidarity with the Ukrainian people and people the world over who actively strive to uphold the most noble principles of academia and of humanity.

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
