# Peer review of "Phase transitions above the upper critical dimension"

_SciPost Physics Lecture Notes_

## Round 1 · Referee Report · Anonymous (Referee 1) · 2022-5-3

Strengths

1- Nicely written

Weaknesses

2- None

Report

The manuscript gives a nice and pedagogical introduction to the critical behavior of systems above the upper critical dimension. I suggest publication.

I have a few minor comments that might improve the presentation.

Eq. (18): it should be stated that the exponential behavior holds for
|x| large (more precisely, to define xi one takes the limit log g(x)/log |x| for |x| to infty). Moreover, a power-correction |x|^p is also expected, i.e. g sim |x|^p exp (-|x|/xi)

Eq. (19): At the first reading I did not understand the meaning of Fourier modes psi_k. I suggest to introduce first Eq. (22) [possibly also Eq. (24)] and then Eq. (19).

Footnote 2, at p. 7: I do not agree on what is stated in the footnote.
In the RG framework, T-Tc, beta-betac or the Campbell proposals involving beta^2, are conceptualy all equivalent. Indeed, the correct variable is a nonlinear scaling field, which is an analytic function of T or beta that
vanishes at the critical point. Which combination (T-Tc or beta-betac)
provides a better approximation to the nonlinear scaling field is a nonuniversal issue. As such, the answer may well be system dependent. So, the statement the "beta-betac" should be preferred in numerical estimates cannot be justified. I would delete the footnote.

Before Eq. (70). To help the reader, note that chi is the derivative of phi0 with respect to the magnetic field.

Argument, eq. (87)-(89). To help the reader, expand the argument, indicating that if d satisfies the bound (89), then <phi^2> sim <phi>^2, i.e. fluctuations can be neglected. It may be helpful, to report again the
definition of chi in Eq. (87).

Before Eq. (91). It would help to write in the text x_phi=d/2-1, just before the equation.

Just below eq. (119). The behavior x^omega holds FOR x LARGE (add in the text).

Just before eq. (121). The author refer to Eq. (40), but this is an equation valid in the infinite volume limit. Therefore, I would rewrite the argument. First, one needs to extend Eq.(40) to a finite volume, which means adding another argument (L/b). Then, by setting b = L, one would obtain (121).

Before Eq. (122): add that the MFT exponent are reported in Table 2.

Last paragraph p. 21: In the third line, Table 1 is quoted. Is this the correct table? Moreover, at the beginning the authors say "it does not seem to be solved by Fisher's DIV mechanism".
I did not understand where this conclusion comes from.

p. 31: the authors should indicate that the ancient Greek character is a "koppa". I am not sure this character is familiar among physicists.

Table 11. It is difficult to understand the relevance of this table in this review paper.

Requested changes

See report

---

## Round 1 · Referee Report · Anonymous (Referee 2) · 2022-6-6

Report

In critical phenomena the upper critical dimension d_uc is the spatial dimensionality above which mean field critical exponents hold. It is well known that, although above d_uc the critical point is Gaussian, basic Gaussian scaling does not reproduce all the mean field exponents. Fisher explained long ago how to overcome the discrepancy observing that, while the field that below d_uc leads to the nontrivial fixed point becomes irrelevant above d_uc, it remains “dangerous” in the sense that it cannot be set to zero without producing the divergence of some scaling function. This settles the case for systems occupying the whole d-dimensional space and, in principle, shows the way for dealing with finite size scaling above d_uc. In practice, however, the case of finite size is seriously complicated by issues like the need to deal with a pseudo-critical point, the role of corrections to scaling, and that of the different types of boundary conditions. These features, together with the fact that the dimensions above d_uc are not experimentally accessible, explain that the problem has mainly been considered by experts of numerical simulations. The present paper provides a historical survey of the developments on this problem, as well as the view of the authors on possible scenarios for points that remain open. It will represent an updated bookkeeping on the subject and I can suggest publication in the Lecture Notes section of SciPost.

---

## Editorial Decision

resubmitted